

# Braincase and endocranial anatomy of two thalattosuchian crocodylomorphs and their relevance in understanding their adaptations to the marine environment

Yanina Herrera[1], Juan Martín Leardi[2,3] and Marta S. Fernández[1]

[1] CONICET. División Paleontología Vertebrados, Museo de La Plata, Facultad de Ciencias Naturales y Museo, Universidad Nacional de La Plata, La Plata, Buenos Aires, Argentina
[2] CONICET. Instituto de Estudios Andinos "Don Pablo Groeber" (IDEAN), Facultad de Ciencias Exactas y Naturales, Departamento de Ciencias Geológicas, Universidad de Buenos Aires, Buenos Aires, Argentina
[3] Universidad de Buenos Aires, Facultad de Ciencias Exactas y Naturales, Facultad de Ciencias Exactas y Naturales, Departamento de Biodiversidad y Biología Experimental, Universidad de Buenos Aires, Argentina

Corresponding author
Yanina Herrera,
yaninah@fcnym.unlp.edu.ar

## ABSTRACT

Thalattosuchians are a group of Mesozoic crocodylomorphs known from aquatic deposits of the Early Jurassic–Early Cretaceous that comprises two main lineages of almost exclusively marine forms, Teleosauridae and Metriorhynchoidea. Teleosaurids were found in shallow marine, brackish and freshwater deposits, and have been characterized as semiaquatic near-shore forms, whereas metriorhynchids are a lineage of fully pelagic forms, supported by a large set of morphological characters of the skull and postcranial anatomy. Recent contributions on Thalattosuchia have been focused on the study of the endocranial anatomy. This newly available information provides novel evidence to suggest adaptations on the neuroanatomy, senses organs, vasculature, and behavioral evolution of these crocodylomorphs. However, is still not clear if the major morphological differences between teleosaurids and metriorhynchids were also mirrored by changes in the braincase and endocranial anatomy. Based on X-ray CT scanning and digital endocast reconstructions we describe the braincase and endocranial anatomy of two well-preserved specimens of Thalattosuchia, the semiaquatic teleosaurid *Steneosaurus bollensis* and the pelagic metriorhynchid *Cricosaurus araucanensis*. We propose that some morphological traits, such as: an enlarged foramen for the internal carotid artery, a carotid foramen ventral to the occipital condyle, a single CN XII foramen, absence of brain flexures, well-developed cephalic vascular system, lack of subtympanic foramina and the reduction of the paratympanic sinus system, are distinctive features of Thalattosuchia. It has been previously suggested that the enlarged foramen for the internal carotid artery, the absence of brain flexures, and the hypertrophied cephalic vascular system were synapomorphies of Metriorhynchidae; however, new information revealed that all of these features were already established at the base of Thalattosuchia and might have been exapted later on their evolutionary history. Also, we recognized some differences within Thalattosuchia that previously have not been received attention or even were overlooked (e.g., circular/bilobate trigeminal foramen, single/double CN XII foramen, separation of the cranioquadrate canal from the external otic aperture,

absence/presence of lateral pharyngeal foramen). The functional significances of these traits are still unclear. Extending the sampling to other Thalattosuchia will help to test the timing of acquisition and distribution of these morphological modifications among the whole lineage. Also comparison with extant marine tetrapods (including physiological information) will be crucial to understand if some (and/or which) of the morphological peculiarities of thalattosuchian braincases are products of directional natural selection resulting in a fully adaptation to a nektonic life style.

## INTRODUCTION

During the Mesozoic, several groups of reptiles displayed secondary adaptations to life in marine environments and some of them were especially successful and thrived as major predators in the sea (*Massare, 1988*; *Mazin, 2001*; *Bardet et al., 2014*). The most taxonomically diverse groups of Mesozoic marine reptiles are Sauropterygia, Ichthyosauria, Squamata, and Testudinata (*Bardet et al., 2014*). Remains of thalattosuchian crocodylomorphs are also abundant and taxonomically diverse in the fossil record; however, this group has received less attention in the scientific literature despite being an important component of the marine vertebrate fauna during the Mesozoic era.

Thalattosuchians are known from aquatic deposits of the Early Jurassic through the Early Cretaceous distributed mainly in the Tethys and Pacific oceans, that comprises two main lineages of almost exclusively marine forms, Teleosauridae and Metriorhynchoidea (*Fraas, 1902*; *Andrews, 1913*; *Jouve, 2009*; *Pol & Gasparini, 2009*; *Young et al., 2010*; *Wilberg, 2015a*). Teleosaurids were recovered in shallow marine, brackish and even freshwater deposits and based on their morphology have been characterized as semiaquatic and near-shore forms (*Hua & De Buffrenil, 1996*; *Martin et al., 2016*; *Johnson et al., 2017*). On the other hand, Metriorhynchidae are a lineage of fully pelagic forms, with a large set of morphological traits related to a life in an open ocean environment (*Fraas, 1902*; *Andrews, 1913*; *Young et al., 2010*).

Among thalattosuchians, the body plan of teleosaurids (elongate and tubular snout, high tooth count, dorsally directed orbits) has been considered as analogous to modern gavials (*Andrews, 1913*; *Westphal, 1962*). On the other hand, the derived morphological features of Metriorhynchidae (i.e., laterally directed orbits, reduced and paddle-like forelimbs, hypocercal tail, strongly ventrally directed sacral ribs, loss of osteoderms, hypertrophied nasal glands for salt excretion, reduced olfactory bulbs and olfactory nasal region, and probably were bearing live young; *Fraas, 1902*; *Andrews, 1913*; *Fernández & Gasparini, 2008*; *Young et al., 2010*; *Herrera, Fernández & Gasparini, 2013*; *Herrera et al., 2017*) confer a unique and easily recognizable body plan that differs from that of typical crocodylomorphs. These morphological and physiological modifications were key-features for the successful invasion of the marine realm.

Until recently, endocranial anatomy of thalattosuchians was poorly known and mainly based on artificial or natural brain endocasts (*Wharton, 2000*; *Herrera, 2015*; *Herrera & Vennari, 2015*). Recent contributions on Thalattosuchia have been focused on the study of the braincase and endocranial anatomy of three-dimensional preserved specimens, based on X-ray computed tomography scanning and 3D visualization techniques. This newly available information provides novel evidence to suggest adaptations of the neuroanatomy, sense organs, vasculature, and behavioral evolution of these crocodylomorphs (see *Fernández et al., 2011*; *Herrera, Fernández & Gasparini, 2013*; *Brusatte et al., 2016*; *Pierce, Williams & Benson, 2017*). However, it is still not clear if the major morphological differences between teleosaurids and metriorhynchids were also mirrored by changes in the braincase and endocranial anatomy.

Herein, we describe the braincase and the brain endocast, vasculature, inner ear, and paratympanic pneumatic cavities of two thalattosuchians: the teleosaurid *Steneosaurus bollensis* (*Jaeger, 1828*), and the metriorhynchid *Cricosaurus araucanensis* (*Gasparini & Dellapé, 1976*). We used these specimens as a tool to evaluate the disparity in the braincase and endocranial anatomy between teleosaurids and metriorhynchids, as these represent members of the two distinct clades of Thalattosuchia. In this sense, we explore whether peculiarities in the braincase and endocranial structures of metriorhynchids correspond to novel traits of this clade or if they were widespread among thalattosuchians. Finally, the significance of these structures for our understanding of the paleobiology of these marine crocodylomorphs will be evaluated.

## MATERIALS AND METHODS

BSPG 1984 I258, referred to as *Steneosaurus bollensis*, was recovered from Toarcian outcrops located in the surroundings of Altdorf (Mittelfranken, Bayern, Germany) and consists of the braincase three-dimensionally preserved with no evidence of post-mortem deformation. It is almost complete except for the most anterior portion of the frontal and the supraoccipital (Figs. 1 and 2). It was X-ray micro-CT scanned in 2015 in a Nanotom Scan, located at the Zoologische Staatsammlung München (Bavaria State Collection of Zoology, Munich, Germany). The dataset consisted of 1,798 slices (2,261 × 2,443 × 1,798 voxel, 0.043 mm voxel size). Due to poor preservation of the external sutures, the bones were segmented separately (Fig. 2).

The holotype of *C. araucanensis* (MLP 72-IV-7-1), recovered from Portada Covunco Member (middle Tithonian) of the Vaca Muerta Fm. exposed at Cerro Lotena (northwestern Patagonia, Argentina), consists of an almost complete three-dimensionally preserved skull. There is no conspicuous evidence of post-mortem deformation except for the slight displacement of the palatines and pterygoids. For the purpose of this contribution we only provide the description of the braincase. MLP 72-IV-7-1 was helically scanned in a X-ray medical CT-scanner in 2007. The data consisted of 471 slices with a pixel size of 0.448 × 0.448 mm, a slice thickness of two mm, a slice increment of 0.999 mm and they were output from the scanner in DICOM format. We are aware that the low CT resolution precludes reconstructing fine three-dimensional models of most cranial nerves, endosseous labyrinth of the inner ear and some regions of the

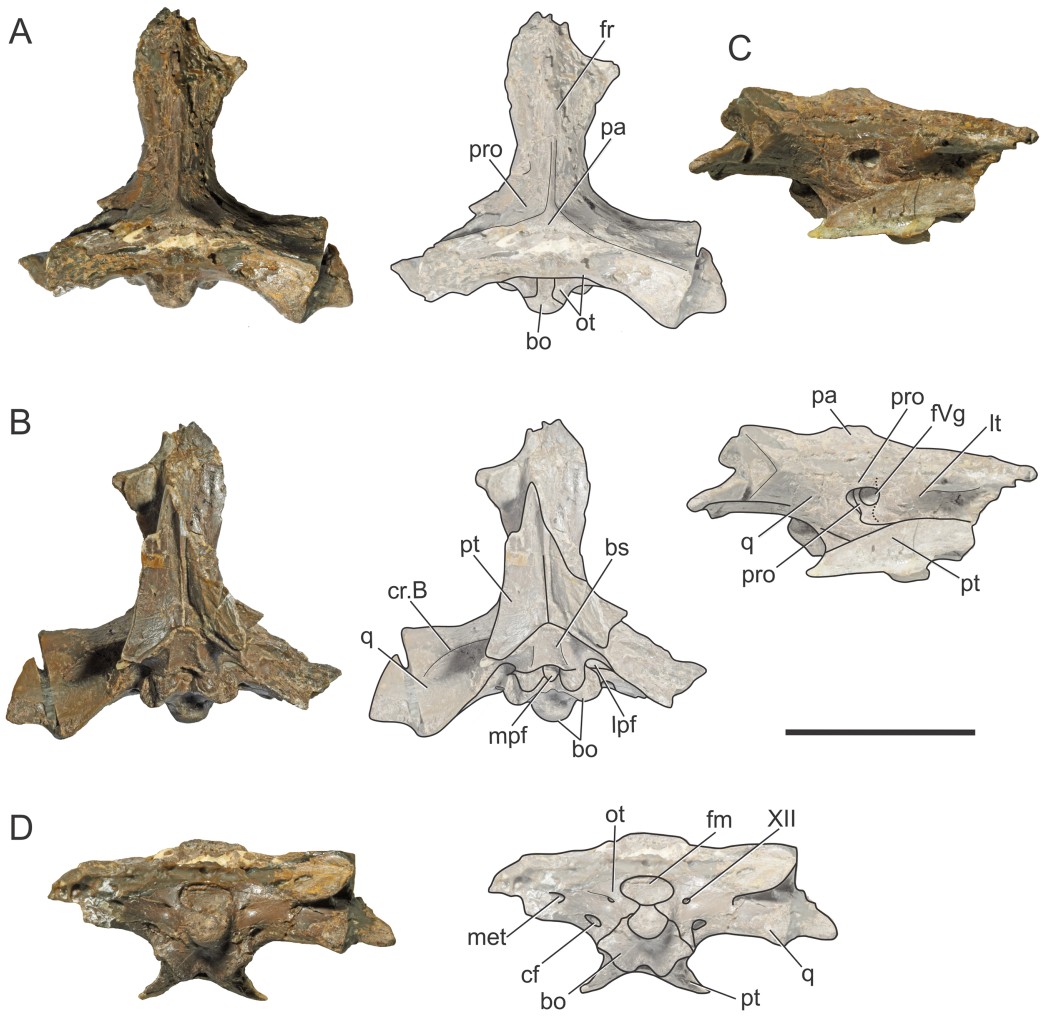

**Figure 1 Braincase of *S. bollensis* (BSPG 1984 I258).** In (A) dorsal; (B) ventral; (C) right anterolateral; and (D) posterior views. Abbreviations: bo, basioccipital; bs, basisphenoid; cf, carotid foramen; cr.B, crest B; fm, foramen magnum; fr, frontal; fVg, foramen for the trigeminal ganglion; lpf, lateral pharyngeal foramen; lt, laterosphenoid; met, metotic foramen; mpf, median pharyngeal foramen; ot, otoccipital; pa, parietal; pro, prootic; pt, pterygoid; q, quadrate; XII, hypoglossal foramen. Dotted white line in (C) shows laterosphenoid-prootic suture, continuous lines show the trigeminal foramen and fossa. Scale bars equal five cm.

paratympanic sinus system. New CT scans of other specimens of *C. araucanensis* will surely be useful to study the internal anatomy with greater detail. However, the general features of the encephalum and the pneumatic cavities can be observed and discussed. The respective CT data files were imported as DICOM files into Materialise Mimics 10.01 (Materialise Inc., Leuven, Belgium) for image segmentation and digital reconstruction.

## RESULTS

### Braincase anatomy of *S. bollensis* (BSPG 1984 I258)

The frontal, squamosals and supraoccipital are not well preserved or are largely incomplete (Figs. 1 and 2), and thus we do not include the description of these bones.

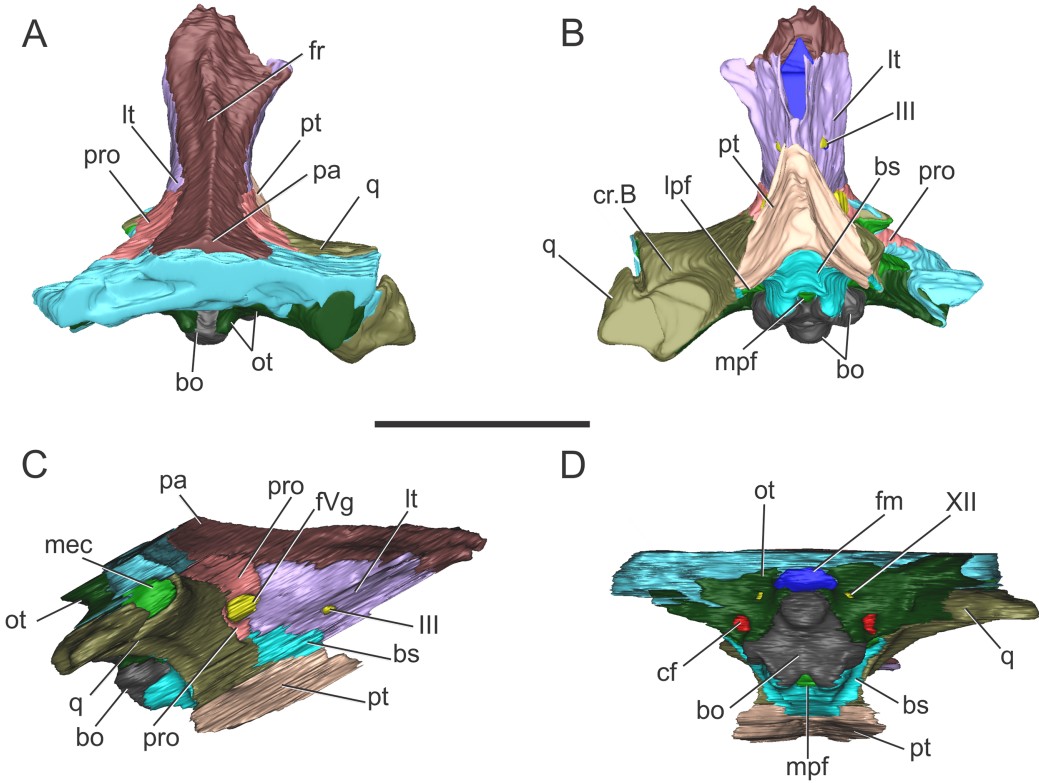

**Figure 2 Reconstruction of the braincase of *S. bollensis* (BSPG 1984 I258).** In (A) dorsal; (B) ventral; (C) right lateral; and (D) posterior views. Abbreviations: bo, basioccipital; bs, basisphenoid; cf, carotid foramen; cr.B, crest B; fm, foramen magnum; fr, frontal; fVg, foramen for the trigeminal ganglion; lpf, lateral pharyngeal foramen; lt, laterosphenoid; mec, middle ear cavity; mpf, median pharyngeal foramen; ot, otoccipital; pa, parietal; pro, prootic; pt, pterygoid; q, quadrate; III, oculomotor foramen; XII, hypoglossal foramen. Scale bar equals five cm.

**Parietal.** The parietal is a single element, as in derived crocodylomorphs (*Sphenosuchus* and more derived taxa) and crocodyliforms (*Clark et al., 2004*; *Leardi, Pol & Clark, 2017*). The parietal is partially preserved (Figs. 1A and 2A). It forms part of the dorsal and posterolateral walls of the braincase. Anteriorly, it contacts the frontal through its elongate anterior process; the dorsalmost region of the intertemporal bar is narrow, forming a sagittal crest (Figs. 1A and 2A), as in all thalattosuchians. Anteroventrally the parietal has a broad contact with the laterosphenoid and posteroventrally with the prootic. The parietal forms the posterior region of the intertemporal bar and the medial and posteromedial margins of the supratemporal fenestrae and fossae (Figs. 1A, 1C, 2A and 2C). Posteriorly, the "parietal table" (*sensu Brusatte et al., 2016*), although slightly incomplete in its posterior region, is less anteroposteriorly developed than in *Steneosaurus* cf. *gracilirostris*. In BSPG 1984 I258 the anterior end of the "parietal table" is almost at the level of the posterior margin of the supratemporal fossa and not at the same plane as the laterosphenoid-prootic suture as in *Steneosaurus* cf. *gracilirostris* (*Brusatte et al., 2016*).

**Prootic.** Both prootics are incompletely preserved. It is exposed on the posteromedial region of the supratemporal fossa (Figs. 1A, 1C, 2A and 2C), as in most thalattosuchians and non-crocodyliform crocodylomorphs (*Clark, 1986*; *Leardi, Pol & Clark, 2017*), and

has a dorsal contact with the parietal; this suture could not be recognized externally but it is recognizable in the CT data (Figs. 2A and 2C). Anteriorly, the prootic contacts the laterosphenoid, and posteriorly the quadrate. The prootic forms the dorsal, posterior and ventral margins of the circular trigeminal foramen, and the dorsal and posterior margins of the trigeminal fossa (Figs. 1C and 2C). A circular trigeminal foramen is also present in *Steneosaurus* cf. *gracilirostris* (NHMUK PV OR 33095), *Machimosaurus buffetauti* (SMNS 91415), and likely in *Teleosaurus cadomensis* (*Jouve, 2009*: Fig. 2). In the posterior margin of the trigeminal foramen the prootic is a slender rod that runs dorsally and separates the trigeminal foramen from the middle ear cavity. Ventral to the trigeminal foramen the prootic contacts anteriorly the laterosphenoid and posteriorly the quadrate (Figs. 1C and 2C).

**Laterosphenoid.** Both laterosphenoids are incompletely preserved (Figs. 1C, 2B and 2C). They form most of the lateral wall of the braincase, and contact posteriorly the prootic, and anteriorly delimits the exit for the olfactory tract. The laterosphenoid forms the anterior margin of the trigeminal fossa and foramen. The laterosphenoid-prootic suture is located at the level of the anterior margin of the trigeminal foramen, and the laterosphenoid does not participate in the dorsal and ventral margins of this foramen, as it only reaches the anterior border of the trigeminal foramen. The trigeminal fossa is not developed anteriorly, thus the laterosphenoid is not excavated (Figs. 1C and 2C). CT data shows that the laterosphenoid contacts the basisphenoid on the floor of the endocranial cavity. Level with the ventral margin of the trigeminal foramen, a groove excavates the lateral surface of the laterosphenoid, which is dorsally delimited by a subtle ridge. This groove is interpreted as the osteological correlate of the ophthalmic branch of the trigeminal nerve (CN $V_1$) (Fig. 1C). In thalattosuchians the presence of a ridge on the region where the laterosphenoid-prootic suture is located has been previously recognized as a unique trait (*Holliday & Witmer, 2009*; *Fernández et al., 2011*), however, in this specimen the ridge is not conspicuous and not tightly in contact with the prootic as in other thalattosuchians.

**Quadrate.** Both quadrates are incompletely preserved, with the right one more complete than the left, and only missing the distal ends of the condyles for the articular (Fig. 1). The ventral aspect of the left quadrate is eroded exposing a concave surface that corresponds to the middle ear cavity. The quadrate contacts dorsomedially the prootic and ventromedially the basisphenoid and the pterygoid (Figs. 1B, 1C, 2B and 2C). The right orbital process of the quadrate is partially covered by sediment and the left one is not preserved (Fig. 1C); however, it appears to not be firmly sutured to the braincase, as in other thalattosuchians (*Machimosaurus buffetauti*, SMNS 91415; *Jouve, 2009*; *Holliday & Witmer, 2009*; *Fernández et al., 2011*; *Herrera, Gasparini & Fernández, 2015*; *Wilberg, 2015a*). The trigeminal fossa is developed posterior to the trigeminal foramen and excavates the anterolateral surface of the quadrate (Figs. 1C and 2C), like in *Steneosaurus pictaviensis* (LPP.M.37), *Machimosaurus buffetauti* (SMNS 91415), *C. araucanensis* (MLP 72-IV-7-1), and "*Metriorhynchus*" cf. *westermanni* (*Fernández et al., 2011*). The extension of the fossa in BSPG 1984 I258, posterior to the trigeminal foramen, is probably exaggerated because this region is damaged. The quadrate does not participate in the

margin of the trigeminal foramen (Fig. 2C). In ventral view, the quadrate contacts the basisphenoid through a serrated suture (Fig. 1B). The quadrate does not reach the basal tuberosities of the basioccipital (Figs. 1B and 2B), unlike in *Steneosaurus* cf. *gracilirostris* (*Brusatte et al., 2016*), and *Pelagosaurus typus* (BSPG 1890 I5, NHMUK PV OR 32599). In BSPG 1984 I258 the main body of the quadrate has a more lateral direction in comparison with other thalattosuchians and forms an angle of about 70° with the sagittal plane of the skull (Fig. 1B), similar to *Steneosaurus edwarsi* (NHMUK PV R 3701). In other thalattosuchians (e.g., *Steneosaurus* cf. *gracilirostris*, NHMUK PV OR 33095; *Pelagosaurus typus*, BSPG 1890 I5, NHMUK PV OR 32599; *Peipehsuchus teleorhinus*, IVPP V 10098; *C. araucanensis*, MLP 72-IV-7-1) this angle is more acute and results in the quadrate's body being more posterolaterally directed. On the ventral surface of the quadrate the "crest B" (*Iordansky, 1973*) marks the origin of the *M. adductor mandibulae posterior* as in most crocodyliforms (Figs. 1B and 2B). In BSPG 1984 I258 "crest B" is sharp and has its medialmost branch posteriorly curved, and it delimits a conspicuous fossa on the ventral surface of the quadrate within the adductor chamber, as in *Steneosaurus edwarsi* (NHMUK PV R 3701). A well-developed "crest B" has also been described in other thalattosuchians (*Jouve, 2009*; *Holliday & Witmer, 2009*; *Fernández et al., 2011*; *Young et al., 2012*; *Herrera, Gasparini & Fernández, 2013*; *Herrera, Gasparini & Fernández, 2015*; *Brusatte et al., 2016*). Due to preservation, the sutures with the otoccipital, and the region of the cranioquadrate foramen could not be described.

**Otoccipitals.** The exoccipitals and opisthotics are fused in a single element, the otoccipital (*Clark, 1986*). Both otoccipitals are incomplete, not well preserved and partially reconstructed (Figs. 1D and 2D). The otoccipital forms the lateral margins of the foramen magnum, but it is not possible to determine if it has some degree of participation in the dorsal margin. The foramen magnum is oval, with the major axis mediolaterally oriented. Also, the otoccipital participates in the dorsolateral region of the occipital condyle contacting the basioccipital, as in most crocodylomorphs (Figs. 1D and 2D). The right paroccipital process is almost complete and is slightly dorsally directed (Figs. 1D and 2D). The ventrolateral flange of the otoccipital contacts the quadrate in the ventral margin of the occipital surface of the skull, lateral to the lateral pharyngeal foramen. The otoccipital forms approximately half of the posterior margin of the lateral pharyngeal foramen (Figs. 1B and 2B). Lateral to the foramen magnum and at the same level with its ventral margin, the single foramen for the passage of the cranial nerve XII (i.e., hypoglossal foramen) is present (Figs. 1D and 2D), like in *C. araucanensis* (MLP 72-IV-7-1), *Teleosaurus cadomensis* (*Jouve, 2009*), "*Metriorhynchus*" cf. *westermanni* (*Fernández et al., 2011*: Fig. 1C) and *Steneosaurus* cf. *gracilirostris* (*Brusatte et al., 2016*), among others, and unlike *Pelagosaurus typus* (BSPG 1890 I5), the metriorhynchid specimen LPP.M.23 and likely "*Metriorhynchus*" *brachyrhynchus* (LPP.M.22), where the CN XII has two foramina on the occipital surface. Approximately at the same level, on the ventrolateral region of the paroccipital process and lateral to the hypoglossal foramen, there is a large foramen. In BSPG 1984 I258 this foramen is interpreted as the common passage of the cranial nerves IX, X, XI and associated vessels (i.e., metotic foramen), as in *Purranisaurus potens* (MJCM PV 2060). It is smaller than the internal carotid artery foramen,

which is conspicuous in this specimen (Figs. 1D and 2D), as in other thalattosuchians. An enlarged foramen for the internal carotid artery was previously proposed as a synapomorphy of Metriorhynchidae (*Pol & Gasparini, 2009*). However, a large or wide internal carotid foramen is also present in the metriorhynchoids *Pelagosaurus typus* (e.g., BSPG 1890 I5), and *Zoneait nargorum* (*Wilberg, 2015a*), and in teleosaurid specimens (e.g., *S. pictaviensis*, LPP.M.37; ?*Steneosaurus* sp., SMNS 59558; *Jouve, 2009*), as mentioned by *Brusatte et al. (2016)*. The internal carotid artery foramen in BSPG 1984 I258 is situated ventrally in the occipital surface, lateral to the basioccipital tuberosities and piercing the otoccipital in a posteroventral direction (Figs. 1D and 2D). A posteroventral or ventrolateral direction of the internal carotid artery foramen is also present in ?*Steneosaurus* sp. (SMNS 59558), *S. pictaviensis* (LPP.M.37), *S. leedsi* (NHMUK PV R 3320), *S. larteti* (GPIT 07283), *M. buffetauti* (SMNS 91415), *T. cadomensis* (*Jouve, 2009*), and differing from the condition in metriorhynchoids, in which the foramen has a strictly posterior direction being only visible in occipital view (see below). Indeed, in some teleosaurid specimens this foramen is only exposed in ventral view (e.g., *S. leedsi*, NHMUK PV R 3320; *Peipehsuchus teleorhinus*, IVPP V 10098; *M. buffetauti*, SMNS 91415).

**Basioccipital.** It forms most of the occipital condyle because the otoccipital participates solely on the dorsolateral region of the condyle, as in *Pelagosaurus typus* (BSPG 1890 I5), *T. cadomensis* (*Jouve, 2009*), *S.* cf. *gracilirostris* (*Brusatte et al., 2016*), and the metriorhynchid specimen from Mörnsheim Formation (BSPG 1973 I195), among others. It forms part of the ventral margin of the foramen magnum. In occipital view, the basioccipital is sutured dorsolaterally and laterally to the otoccipital (Figs. 1D and 2D). In ventral view, the basioccipital contacts anteriorly the basisphenoid. The basioccipital forms most of the basioccipital tuberosities, with its posterolateral region located more dorsally than its medial region. On the anteromedial surface of the tuberosities, it is sutured to the basisphenoid. Between the tuberosities this bone forms the posterior margin of the median pharyngeal foramen. Also, the basioccipital forms roughly half of the posterior margin of the lateral pharyngeal foramen (Figs. 1B and 2B).

**Basisphenoid.** The basisphenoid is widely exposed in ventral view. Anteriorly it contacts the pterygoids through a "V"-shaped suture with the apex anteriorly directed, the quadrates laterally, and posteriorly the basioccipital, forming the anterior and lateral margins of the median pharyngeal foramen. The anteroventral surface of the basisphenoid bears two anteroposteriorly directed crests separated by a concave surface. This anteroventral surface is wider than the median pharyngeal foramen (Figs. 1B and 2B), a similar condition as the one present in *S. bollensis* (SMNS 15951b, SMNS 15816). The basisphenoid has two posterolateral processes that form the anterior margin of the lateral pharyngeal foramen; these processes contact laterally the quadrates through a serrated suture (Fig. 1B). The presence of the lateral pharyngeal (Eustachian) foramina in *S. bollensis* is shared with other teleosaurids (e.g., *Peipehsuchus teleorhinus*, IVPP V 10098; *S. pictaviensis*, LPP.M.37; *T. cadomensis*, *Jouve, 2009*) and basal metriorhynchoids (*Pelagosaurus typus*, BSPG 1890 I5; *Dufeau, 2011*: Figs. 1–6C, D; *Pierce, Williams & Benson, 2017*). However, this contrast with the condition of most metriorhynchids (e.g., *C. araucanensis*; the metriorhynchid specimen from Mörnsheim Formation, BSPG 1973

I195; *Purranisaurus potens*, MJCM PV 2060; *Metriorhynchus superciliosus*, SMNS 10116; "*Metriorhynchus*" *westermanni*, MDA 1) where the lateral pharyngeal foramina are absent. CT data shows that anteriorly, the basisphenoid is sutured dorsally to the laterosphenoid, enclosing the pituitary fossa. On the floor of the endocranial cavity, the internal foramina for the passage of the CN VI were recognized. The canals of these cranial nerves are directed anteroventrally, entering the pituitary fossa dorsal to the internal carotid artery.

## Endocranial anatomy of *S. bollensis* (BSPG 1984 I258)

**Morphology of the brain endocast.** The endocast of the BSPG 1984 I258 comprises the posterior region of the forebrain to the medulla oblongata, lacking the anterior portion of the olfactory tract. Poor preservation of the bones surrounding the brain ventrally resulted in an incomplete reconstruction of the anterior region of the pituitary fossa and of the anteroventral region of the brain (Fig. 3).

The brain shows a subtle anteroposterior differentiation, with the forebrain–midbrain and midbrain–hindbrain flexures not well marked (Figs. 3A and 3B). This feature is also present in other thalattosuchians (*Fernández et al., 2011*; *Herrera, 2015*; *Herrera & Vennari, 2015*; *Brusatte et al., 2016*; *Pierce, Williams & Benson, 2017*). The lateral projection of the bulbous cerebral hemispheres (at the level of the maximum width of cerebrum) is less extended (Figs. 3C and 3D) in comparison with other thalattosuchians (see below) and extant crocodiles (*Witmer et al., 2008*: Fig. 6.3B; *Bona & Paulina Carabajal, 2013*: Fig. 6E; *Bona, Paulina Carabajal & Gasparini, 2017*: Fig. 7A). Although preservation precludes the full reconstruction of the pituitary fossa, its general shape and orientation are discernible (Fig. 3A). The pituitary fossa is anteroposteriorly elongated, as in other thalattosuchians (*Brusatte et al., 2016*; *Pierce, Williams & Benson, 2017*) while it contrasts with the rather anteroposteriory shorter pituitary fossa of other crocodyliforms (e.g., *Gavialis gangeticus*, *Pierce, Williams & Benson, 2017*; *Simosuchus clarki*, *Kley et al., 2010*; *Sebecus icaeorhinus*, *Colbert, 1946a*). As in most crocodyliforms, the pituitary fossa in BSPG 1984 I258 is posteroventrally projected, having its anterior region more dorsally positioned than its posterior one. Posteriorly, the extension of the pituitary fossa reaches the level of the posterior margin of the trigeminal foramen (Fig. 3A).

**Vascular elements.** The rostral and caudal middle cerebral veins and the internal carotid artery were identified. The rostral middle cerebral vein forms a swelling on the dorsal region of the endocast, posteriorly to the cerebral hemispheres and dorsally to the trigeminal nerve (CN V) (Fig. 3B). The rostral middle cerebral vein exits the braincase through the trigeminal foramen (Figs. 3B and 3D), as in other thalattosuchians. The dorsal longitudinal sinus, that overlays the brain (Fig. 3D), is not as ridge-like as in *C. araucanensis* (see below), because the dorsal portion of the endocast is flat (Figs. 3A and 3B). Posterodorsally on the endocast, the roots of the caudal middle cerebral vein were reconstructed. These veins are dorsolaterally projected and then turn laterally, within the temporal canal, toward the cranioquadrate canal (Figs. 3A and 3D). These structures were previously recognized in other thalattosuchians and identified as related to vascular elements under different names: portion of the dorsal venous sinus system (*Wharton, 2000*), "cavity 1" (*Fernández et al., 2011*), posterior portion of the transverse

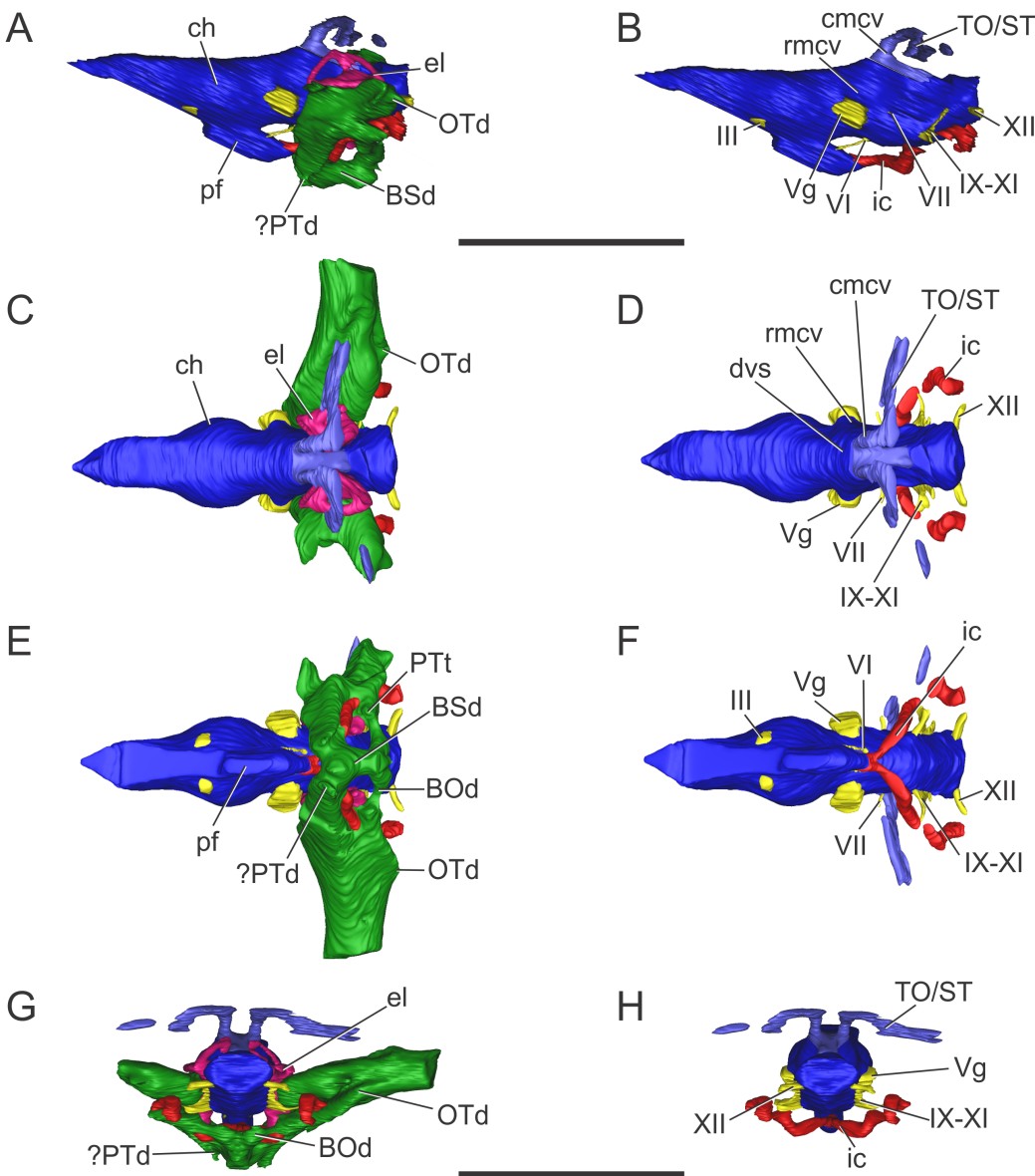

**Figure 3 Endocranial anatomy of S. bollensis (BSPG 1984 I258).** In (A and B) lateral; (C and D) dorsal; (E and F) ventral; and (G and H) posterior views. Abbreviations: BOd, basioccipital diverticulum; BSd, basisphenoid diverticulum; ch, cerebral hemisphere; cmcv, caudal middle cerebral vein; dvs, dorsal venous sinus; el, endosseous labyrinth of the inner ear; ic, internal carotid artery; OTd, otoccipital diverticulum; pf, pituitary; ?PTd, ?pterygoid diverticulum; PTt, pharyngotympanic tube; rmcv, rostral middle cerebral vein; TO/ST, temporo-orbital/stapedial vein; Vg, trigeminal ganglion; III, VI, VII, IX–XI, XII, cranial nerves. Scale bars equal five cm.

sinus/posterior middle cerebral vein (*Brusatte et al., 2016*); or branches of the dorsal longitudinal sinus (*Pierce, Williams & Benson, 2017*). Due to poor preservation, the distal part of this vascular canal (temporal canal) its relationship with the cranioquadrate canal cannot be traced with confidence. However, the preserved morphology is consistent with those of other thalattosuchians, and the lateral projection of the preserved temporal canal suggests that both structures are confluent (Fig. 3G).

The carotid canals run from the ventral region of the occipital surface of the skull to the pituitary fossa (Figs. 3B and 3F). The most posterior portion runs parallel to the midline of the skull for a short way (this portion is shorter than in *C. araucanensis*, see below), to turn obliquely afterward. In BSPG 1984 I258 the longest part of the carotid canal is directed from posterolaterally to anteromedially (Fig. 3F). The carotid canals enter the pituitary fossa piercing its posterior wall through two separate foramina. The carotid canals are not completely ossified, thus the middle portion of the canal could not be segmented separately from the pharyngotympanic sinus (Figs. 3B, 3D and 3F). A similar condition has been recognized in *S.* cf. *gracilirostris* (*Brusatte et al., 2016*) and other extinct crocodylomorphs and extant crocodilians (*Sedlmayr, 2002*; *Bona, Degrange & Fernández, 2013*; *Dufeau & Witmer, 2015*).

**Nerves.** Canals of the III, V, VI, VII, IX–XI, and XII cranial nerves were recognized and reconstructed. As we mentioned above, the anteroventral region of the braincase of BSPG 1984 I258 is damaged in a way that some of the nerves that originate from the ventrolateral region of the midbrain could not be identified.

On the lateral wall of the braincase, ventrolaterally to the cerebral hemispheres, the foramen that pierces the laterosphenoid is here interpreted as the foramen for the oculomotor nerve (CN III) (Figs. 3B and 3F). The large trigeminal foramen which contains the trigeminal ganglion (Vg) is identified as it is ventrolaterally projected from the endocast (Figs. 3B, 3D and 3F).

Cranial nerve VI (abducens) exits the endocranial cavity through the basisphenoid via individual foramina, posteroventrally to CN V (Figs. 3B and 3F). Two passages for the branches of CN VI project slightly anteroventrally from the ventral side of the hindbrain and pass laterally to the pituitary fossa (Fig. 3B). A small canal posterior to the trigeminal nerve (CN V) foramen is identified as the facial nerve canal (CN VII) which exits the endocranial cavity through a foramen in the prootic (Fig. 3B). Cranial nerves IX, X, XI, and XII originate from the lateral region of the hindbrain. Cranial nerves IX, X, XI exit the endocranial cavity through a dorsoventrally elongated metotic foramen. Posteriorly, cranial nerve XII (hypoglossal) has a single root on the endocast and only one external opening in the posterior surface (Figs. 3B, 3D, 3F and 3H).

**Paratympanic sinus system.** In BSPG 1984 I258 several interconnected diverticular expansions from both the pharyngotympanic sinus and median pharyngeal sinus (*sensu Dufeau & Witmer, 2015*) have been identified (Figs. 3A, 3C, 3E and 3G). As the condition present in most crocodyliforms (e.g., *Protosuchus richardsoni*, *Notosuchus terrestris*, *Caiman latirostris*), the paratympanic sinus system of *S. bollensis* communicates with the pharynx via a single medial foramen (median pharyngeal foramen), and two lateral foramina (lateral pharyngeal foramina).

The median pharyngeal tube (= median Eustachian tube) bifurcates in two paired system of pneumatic canals, an anterior pair and a posterior pair (anterior and posterior communicating canals, *sensu Miall, 1878*). The posterior communicating canals diverge almost at 90° from the median canal. They are then directed laterally and dorsally and connect with the middle ear cavity. The posterior communicating canals bear some expansions on their path to contribute to the pneumatization of the posterior part of

the basioccipital (basioccipital diverticulum *sensu Dufeau & Witmer, 2015*) (Figs. 3E and 3G). The anterior communicating canals are connected to the median pharyngeal tube through a wide anteroposteriorly directed tube that runs through the ventral surface of the basisphenoid (anterodorsal branch of the basisphenoid diverticulum) (Figs. 3A and 3E). The anteriormost part of the ventral canal is expanded, in the same region where the two dorsolaterally directed anterior communicating canals originate. At this point a slight ventral projection of the anterior communicating canals is seen, however, it is difficult to evaluate if this pneumatization continues into the pterygoids, forming a pterygoid diverticulum (Figs. 3A, 3E and 3G). The anterior communicating canals are much broader than the posterior ones, and after a short dorsoventral extension, enter the middle ear cavity on its anteroventral region (Fig. 3E).

The middle ear cavity is mediolaterally elongated and tubular, due to the particular thalattosuchian condition where the quadrate extends its limit well laterally when compared to other crocodyliforms. The quadrate lacks any well-developed additional pneumatization (Figs. 3A, 3C, 3E and 3G). That is, BSPG 1984 I258 lacks any infundibular diverticulum, like *Pelagosaurus typus* (*Dufeau, 2011*), *S.* cf. *gracilirostris* (contra *Brusatte et al., 2016*; see below), and *C. araucanensis* (see below). In derived crocodylomorphs (i.e., *Macelognathus*) and most crocodyliforms (e.g., *Protosuchus richardsoni*, *Notosuchus terrestris*, *Caiman latirostris*) the quadrate is heavily pneumatized, both anterior to the otic aperture (infundibular diverticulum) and on the distal body of the quadrate (quadrate diverticulum). These pneumatizations communicate with the middle ear cavity and also have an independent external opening through one subtympanic foramen (or quadrate fenestra) (e.g., *Macelognathus*, *C. latirostris*) or multiple (e.g., *Junggarsuchus*, *P. richardsoni*, *Notosuchus*) (*Leardi, Pol & Clark, 2017*). Thus, the subtympanic foramina are the external osteological correlates of these quadrate pneumatizations (*Dufeau & Witmer, 2015*). Furthermore, no thalattosuchian with the presence of a subtympanic foramen has been reported (e.g., *C. araucanensis*, *S. bollensis*, *Dakosaurus*, *Pelagosaurus*, *Teleosaurus*), reinforcing the interpretation of the lack of quadrate pneumatization in thalattosuchians. In other recent contributions, the infundibular diverticulum has been identified in the teleosaurid *S.* cf. *gracilirostris* (*Brusatte et al., 2016*), however, this specimen does not have any individualized pneumatization that invades the quadrate anteriorly to the otic aperture, and what was identified as the pneumatic inflations of the suspensorium are not separated from the middle ear cavity. The posterior part of the middle ear cavity of BSPG 1984 I258 bears a posterior sheet-like expansion, just at the level where the posterior communicating canal enters the middle ear (Fig. 3G). This posterior laminar expansion slightly pneumatizes the pterygoid process of the quadrate, but it never reaches the distal body of the quadrate, thus not forming a proper quadrate diverticulum (*sensu Dufeau & Witmer, 2015*). This condition is shared with other thalattosuchians, as the absence of a quadrate diverticulum was described previously for *P. typus* (*Dufeau, 2011*) and *S.* cf. *gracilirostris* (*Brusatte et al., 2016*).

Anterodorsally, the paratympanic cavity of BSPG 1984 I258 is convex and slightly projected, and as a result the posteroventral surface of the prootic is concave. A similar morphology has been identified for *S.* cf. *gracilirostris* and *P. typus*, which lead to the

identification of this dorsal projection of the paratympanic cavity as the prootic diverticulum (*Brusatte et al., 2016*; *Pierce, Williams & Benson, 2017*). However, in modern crocodylians the prootic diverticulum is positioned at the level of the semicircular canals, just anterior to them, and dorsal to the trigeminal ganglion (Vg), forming an isolated pneumatic recess (*Dufeau & Witmer, 2015*). None of the features mentioned before can be observed in the CT data of any of the thalattosuchians examined to date. Thus, a well-developed prootic diverticulum seems to be absent in thalattosuchians.

On the other hand, the middle ear cavity has a convex profile in lateral view and, unlike derived non-crocodyliform crocodylomorphs (*Kayentasuchus*, *Dibothrosuchus*, *Junggarsuchus*, *Macelognathus*) and most crocodyliforms (e.g., *Protosuchus richardsoni*, *Caiman latirostris*), it does not invade the prootic. In derived crocodylomorphs a pneumatic cavity has been described in the posterodorsal region of the prootic, usually referred as the mastoid antrum. In crocodyliforms this sinus (intertympanic sinus, *sensu Dufeau & Witmer, 2015*) penetrates into the supraoccipital and passes through it, connecting the middle ear and paratympanic pneumatizations from both sides (*Clark, 1986*). This feature is not present in *S. bollensis* nor in other thalattosuchians where CT data has been made available (*Brusatte et al., 2016*; *Pierce, Williams & Benson, 2017*) or where natural breakage of the supraoccipital has allowed observation of this feature (*Wilberg, 2015b*).

Finally, in BSPG 1984 I258 the pharyngotympanic sinus is expanded posteriorly, partially pneumatizing the otoccipital. The posterior sheet-like expansion of the middle ear cavity mentioned above, before entering the quadrate, expands as it enters the ventral part of the otoccipital (Figs. 3A, 3C, 3E and 3G). The dorsal region of the otoccipital is also pneumatized by a posterior rounded evagination of the pharyngotympanic sinus. However, this pneumatization is restricted to the ventral part of the otoccipital. A similar condition has been reported in other thalattosuchians (*Dufeau, 2011*: Figs. 1–6C, 1–6D; *Brusatte et al., 2016*; *Pierce, Williams & Benson, 2017*).

**Endosseous labyrinth of the inner ear.** The general aspect of the endosseous labyrinth in BSPG 1984 I258 is similar in shape to that of extant and extinct crocodilians (Fig. 4), that is, a triangular vestibular apparatus dorsally and an elongated cochlea ventrally (*Witmer et al., 2008*; *Bona, Degrange & Fernández, 2013*; *Pierce, Williams & Benson, 2017*). The anterior semicircular canal is slightly longer than the posterior one, which is similar to the lateral canal (Figs. 4B and 4D). The cochlear ducts extend largely ventrally, with only a slight medial component (Figs. 4A and 4C). In BSPG 1984 I258, the complete inner ear is approximately 20.5 mm tall and has a maximum width of 16 mm at the level of the semicircular canals.

## Braincase anatomy of *C. araucanensis* (MLP 72-IV-7-1)

**Frontal.** The frontal is completely fused. The postorbital processes form an acute angle of about 45° with the midline of the skull. The anteromedial process of the frontal wedges anteriorly between the posteromedial processes of the nasal and extends further anteriorly than the level of the posterior margin of the preorbital fossa. The frontal has a reduced participation in the dorsal margin of the orbit (Fig. 5A). Posterolaterally, the

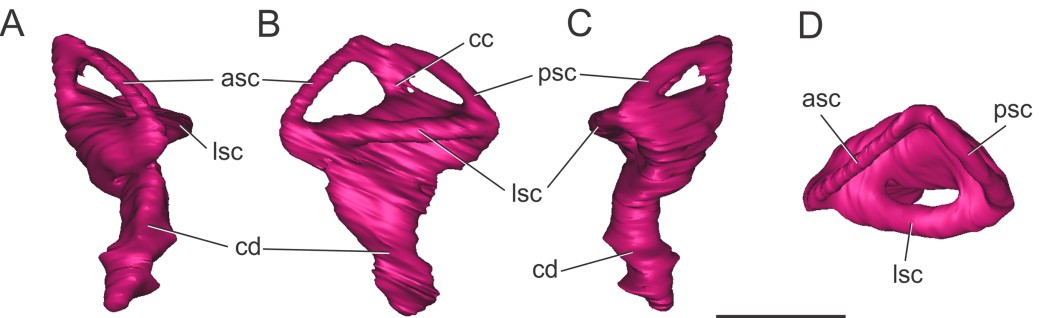

**Figure 4 Left endosseous labyrinth of *S. bollensis* (BSPG 1984 I258).** In (A) anterior; (B) lateral; (C) posterior; and (D) dorsal views. Abbreviations: asc, anterior semicircular canal; cc, common crus; cd, cochlear duct; lsc, lateral semicircular canal; psc, posterior semicircular canal. Scale bar equals one cm.

frontal contacts the postorbital trough a "V"-shaped suture with the apex pointed posteriorly. The postorbital process of the frontal forms the posterodorsal margin of the orbit, and the anteromedial margin of the supratemporal fossa. The frontal extends posteroventrally and forms most of the anterior floor of the supratemporal fossa.

In ventral view the frontal contacts anteriorly with the prefrontal, and posteroventrally, with the laterosphenoid forming the exit for the olfactory tract. The groove on the skull roof is the osteological correlate of the olfactory tract and is delimited laterally by two low *crista cranii*. The anterior portion of the groove is mediolaterally wider than the posterior one.

**Parietal.** The parietal is a "T"-shaped element with an anterior (frontal) process and two lateral (squamosal) processes (Fig. 5A). The anterior process forms the posterior region of the intertemporal bar and contacts the frontal; the lateral process is sutured to the squamosal, both via serrated sutures (Figs. 5A and 6A). Posterodorsally, the parietal bears a large posterior notch between the squamosal processes, forming a semicircular or "U"-shaped structure in dorsal view (Fig. 5A). This posterior parietal notch is also present in other metriorhynchids such as: *Cricosaurus lithographicus* (MOZ-PV 5787), *Cricosaurus elegans* (BSPG AS I 504), "*Metriorhynchus*" *brachyrhynchus* (LPP.B.1), and the metriorhynchid specimen from the Mörnsheim Formation (BSPG 1973 I195). The parietal is well-extended posteriorly, having participation in the central region of the occipital surface. Given this condition, the supraoccipital is excluded from the dorsal aspect of the skull (Fig. 5C) as in most non-eusuchian and non-notosuchian crocodyliforms (*Clark, 1986*).

In occipital view the parietal is ventrally sutured to the supraoccipital and ventrolaterally has a reduced contact with the otoccipital near the midline. In the place where the parietal contacts the otoccipital and the supraoccipital, there is a reduced and obliterated fossa, which corresponds topographically to where the posttemporal fenestra is located in other crocodylomorphs. The lateral processes of the parietal develop a rim over the occipital surface, which continues in the squamosal (Fig. 5C).

Within the supratemporal fossa, the parietal is projected ventrally, forming part of the posterolateral wall of the braincase as well the medial margin of the supratemporal fenestra

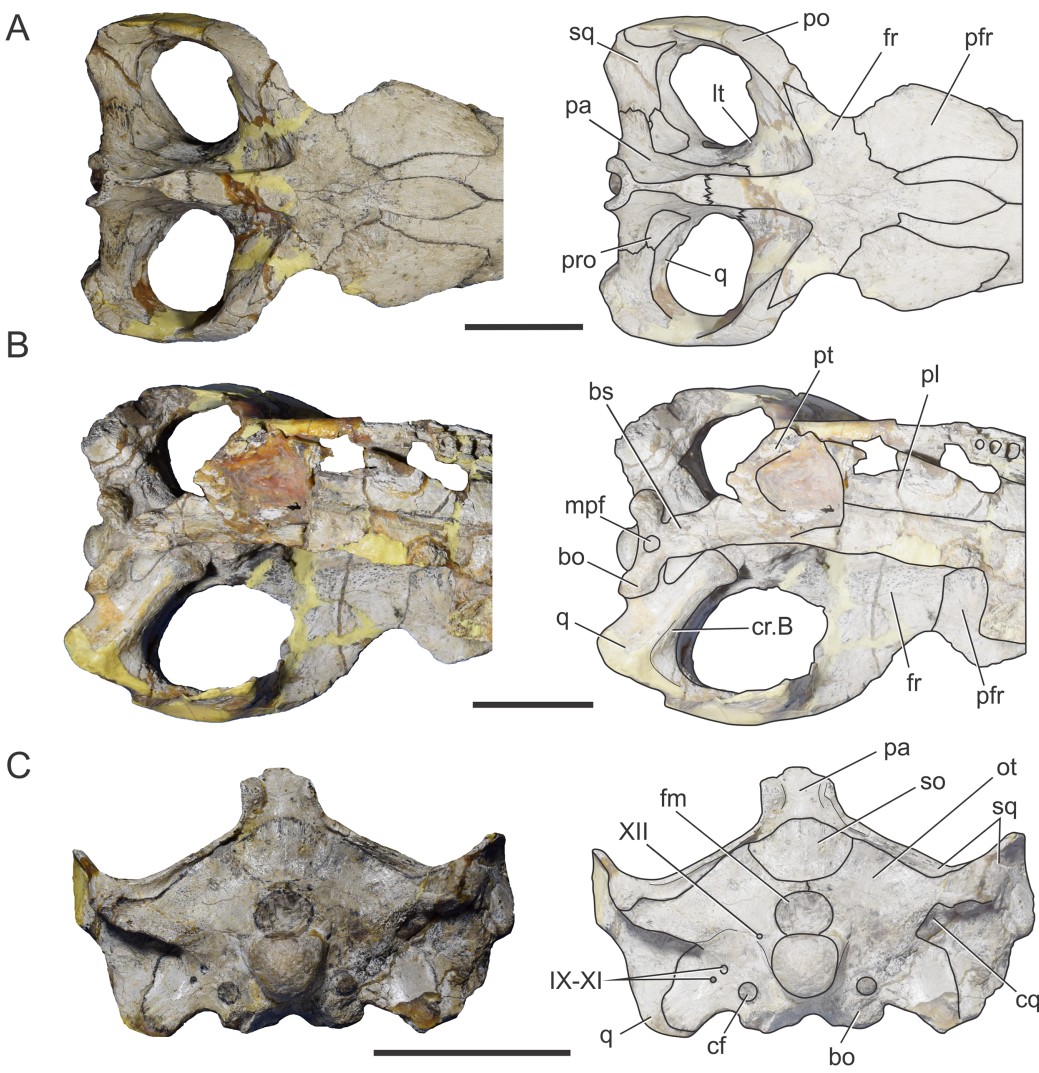

**Figure 5 Braincase of *C. araucanensis* (MLP 72-IV-7-1).** In (A) dorsal; (B) ventral; (C) posterior views. Abbreviations: bo, basioccipital; bs, basisphenoid; cf, carotid foramen; cq, cranioquadrate foramen; cr.B, crest B; fm, foramen magnum; fr, frontal; lt, laterosphenoid; mpf, medial pharyngeal foramen; ot, otoccipital; pa, parietal; pfr, prefrontal; pl, palatine; po, postorbital; pro, prootic; pt, pterygoid; q, quadrate; so, supraoccipital; sq, squamosal; IX–XI, XII, foramina for cranial nerves. Scale bars equal five cm.

(Figs. 5A and 6A). Within the fossa, the parietal is anteriorly sutured to the frontal through a transverse and interdigitated suture, anteroventrally to the laterosphenoid, and posteroventrally to the prootic. The parietal forms the dorsomedial margin of the temporo-orbital foramen (Fig. 6A).

**Squamosal.** The squamosal contributes to the posterior and posterolateral margins of the supratemporal fossa and fenestra. The squamosal participation in the supratemporal arch is reduced, with the postorbital contributing around 75% of the arch. Anterolaterally the squamosal contacts the postorbital, while posteromedially it contacts the parietal (Fig. 5A). The squamosal-postorbital suture is serrated on the lateral aspect of the skull, and within the supratemporal fossa the suture is straight and "V"-shaped with the apex

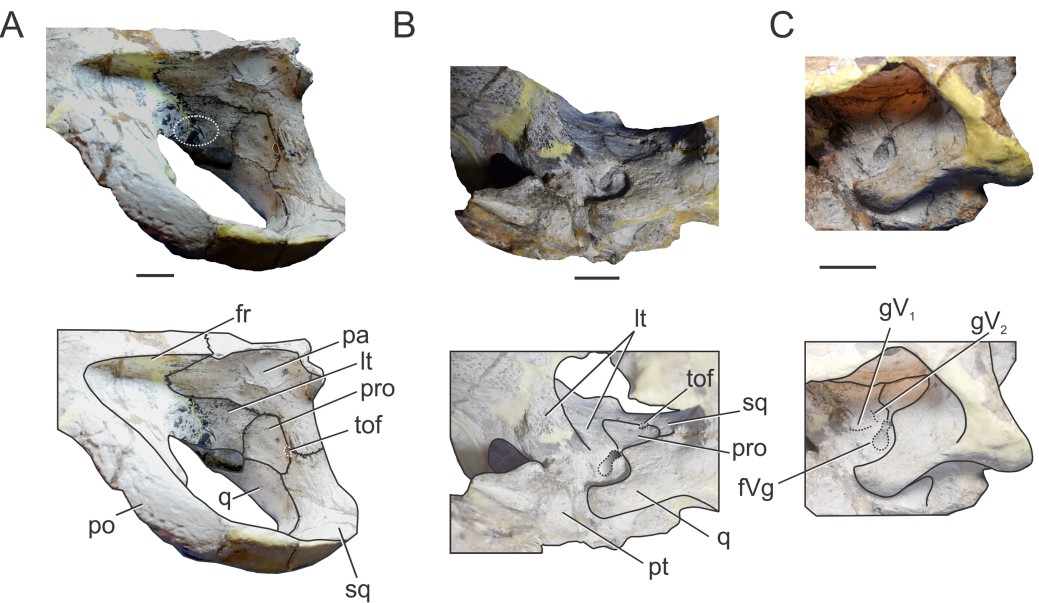

**Figure 6 Braincase of *C. araucanensis* (MLP 72-IV-7-1).** In (A) left dorsolateral; (B) left ventrolateral; and (C) left ventrolateral views. Abbreviations: fr, frontal; fVg, foramen for the trigeminal ganglion; gV₁, groove for the ophthalmic branch of the trigeminal nerve; gV₂, groove for the maxillary branch of the trigeminal nerve; lt, laterosphenoid; pa, parietal; po, postorbital; pro, prootic; pt, pterygoid; q, quadrate; sq, squamosal; tof, temporo-orbital foramen. The white circle on (A) marks the blood vessel infillings that cover the cerebral hemispheres. Scale bars equal two cm.

oriented anteriorly. In dorsal view the squamosal is narrow and slightly concave (Fig. 5A). Within the supratemporal fossa the squamosal extends ventrally contacting the quadrate and the prootic (Fig. 6A). The suture with the latter is given by the ventral branch of the dorsomedial process of the squamosal, which is broader than the dorsal one. The squamosal forms the dorsolateral margin of the reduced temporo-orbital foramen (Fig. 6A).

In occipital view the squamosal is sutured to the dorsolateral margin of the paroccipital process via a rounded posterior process (Fig. 5C). Visible in lateral and posterior views is the smooth and slightly concave subcircular structure of the squamosal present in other thalattosuchians (e.g., *S. leedsi*, NHMUK PV R 3320; *D. andiniensis*, MOZ-PV 6146; *C. lithographicus*, MOZ-PV 5787; *Maledictosuchus riclaensis*, MPZ 2001/130a; *Torvoneustes coryphaeus*, *Young et al., 2013*; *Tyrannoneustes lythrodectikos*, *Foffa & Young, 2014*).

**Prootic.** The prootic is broadly exposed on the lateral wall of the braincase and it is exposed on the posteromedial margin of the supratemporal fossa (Fig. 5A), as in non-crocodyliform crocodylomorphs and other thalattosuchians (*Leardi, Pol & Clark, 2017*). This bone has a subpentagonal shape and contacts anteriorly the laterosphenoid, dorsally the parietal, posteriorly the squamosal, and ventrally the quadrate (Fig. 6A). The prootic forms the ventral margin of the temporo-orbital foramen (Fig. 6A), as in *Pelagosaurus typus* (NHMUK PV OR 32599), *Teleosaurus cadomensis* (*Jouve, 2009*) and *Purranisaurus potens* (MJCM PV 2060). In MLP 72-IV-7-1, the prootic has a reduced contribution to the dorsal margin of the trigeminal fossa (although this feature is variable among the

*C. araucanensis* specimens). The prootic forms the posterior half of the bilobate trigeminal foramen (Figs. 6B and 6C).

**Laterosphenoid.** It forms most of the lateral and anteroventral walls of the endocranial cavity, surrounding the cerebral hemispheres. The laterosphenoid forms the anterior margin of the supratemporal fossa. Anterior and dorsally it contacts the frontal and postorbital (Fig. 5A). Within the supratemporal fossa, the laterosphenoid has a broad dorsal contact with the parietal and briefly contacts the frontal. The laterosphenoid is posteriorly sutured to the prootic through a suture that forms a pronounced ridge (Fig. 6A), as in other thalattosuchians (e.g., *Purranisaurus potens*, MJCM PV 2060; *Metriorhynchus superciliosus*, SMNS 10116; *Pelagosaurus typus*, BSPG 1890 I5; *S. bollensis*, SMNS 15951b; *Holliday & Witmer, 2009*; *Fernández et al., 2011*; *Brusatte et al., 2016*). In ventral view, the anterodorsal region contacts its counterpart and together with the frontal delimits the exit for the olfactory tract. Anteroventral to the trigeminal fossa, the laterosphenoid contacts the pterygoid (Fig. 6B).

The laterosphenoid forms the anterior and anteroventral margins of the trigeminal foramen. In MLP 72-IV-7-1 the trigeminal foramen is bilobate-shaped, with a posterodorsal lobule much smaller than the anteroventral one (Fig. 6B), as in "*Metriorhynchus*" cf. *westermanni* (MDA 2), *Dakosaurus* cf. *andiniensis* (MOZ-PV 089), the metriorhynchid specimen LPP.M.23, and *Pelagosaurus typus* (NHMUK PV OR 32599). The same morphology is observed in the natural casts of the brain of *C. araucanensis* (e.g., MLP 73-II-27-3, MLP 76-II-19-1, MOZ-PV 7261; *Herrera, 2015*; *Herrera & Vennari, 2015*) as two lobules were identified: a small lobule was interpreted as the middle cerebral vein while a large one was assumed to correspond to the trigeminal nerve (CN V). Anterior to the trigeminal foramen the laterosphenoid is slightly excavated, forming a shallow triangular anterior trigeminal fossa (Fig. 6C). This fossa is dorsally delimited by a crest which is interpreted as the osteological correlate of the opthalmic branch of the trigeminal nerve (CN $V_1$), as it was identified in "*M.*" cf. *westermanni* (*Fernández et al., 2011*) and *S.* cf. *gracilirostris* (*Brusatte et al., 2016*). Dorsal to this crest, there is an anterodorsally directed groove that we interpret as the correlate of the maxillary branch of the trigeminal nerve (CN $V_2$) (Fig. 6C).

Anterodorsal to the trigeminal fossa, the left laterosphenoid is eroded leaving exposed the natural cast of the brain, specifically the dorsal region of the cerebral hemispheres. Here, there are two small blood vessel infillings with bone tissue preserved between them (Fig. 6A), as in the dorsal region of the cerebral hemispheres of the metriorhynchids from Vaca Muerta Fm. MOZ-PV 089, MOZ-PV 7201, MOZ-PV 7261 (*Herrera, 2015*; *Herrera & Vennari, 2015*).

**Quadrate.** In MLP 72-IV-7-1 the quadrates are almost completely preserved, lacking only part of the condylar region (Figs. 5B and 6A–6C). Within the supratemporal fossa, the quadrate contacts dorsally the squamosal and medially the prootic. In MLP 72-IV-7-1 the quadrate does not contact the laterosphenoid, as in most thalattosuchians (*Clark, 1986*), because the prootic is exposed in the supratemporal fossa, thus precluding the contact between these two bones (*Leardi, Pol & Clark, 2017*) (Fig. 6A). In MLP 72-IV-7-1 the fossa excavates the quadrate, thus this bone forms the posterior and ventral margins of

the trigeminal fossa. The trigeminal fossa is broadly developed posterior to the trigeminal foramen (Figs. 6B and 6C), as in most metriorhynchids (e.g., "M." brachyrhynchus, LPP. M.22; "M." westermanni and "M." cf. westermanni; Fernández et al., 2011; Plesiosuchus manselli, Young et al., 2012) and some teleosaurids (e.g., Machimosaurus buffetauti, SMNS 91415; S. pictaviensis, LPP.M.37). The quadrate does not participate on the temporo-orbital foramen (Figs. 6A and 6B), as in Pelagosaurus typus (BSPG 1890 I5, NHMUK PV OR 32599), and unlike the condition present in Teleosaurus cadomensis (Jouve, 2009) and Machimosaurus buffetauti (SMNS 91415; Martin & Vincent, 2013), where the quadrate participates very slightly to the ventrolateral margin. The orbital process of the quadrate remains free of bony attachment (Figs. 6B and 6C), as in other thalattosuchians.

In ventral view, the quadrate contacts the basioccipital and the basisphenoid medially, but due to damage on the holotype specimen (MLP 72-IV-7-1), we cannot describe in detail this region as well as the contact with the pterygoids. "Crest B" of the quadrate is low and wide and it is developed on the anterior border of the quadrate (Fig. 5B). Anterior to the "crest B" the quadrate is not exposed (in ventral view), as in "M." cf. westermanni (MDA 2; Fernández et al., 2011: Fig. 1B), Purranisaurus potens (MJCM PV 2060), and the metriorhynchid specimen (LPP.M.23). The pterygoid process of the quadrate is broad and it widens anterodorsally, forming an expanded distal end (Fig. 5B).

In occipital view, the quadrate contacts the ventrolateral flange of the otoccipital. The quadrate forms the lateral margin of the cranioquadrate canal (Fig. 5C). In MLP 72-IV-7-1, the cranioquadrate foramen and canal are separated from the external otic aperture by a bony lamina (Fig. 5C), as in other metriorhynchids (e.g., Purranisaurus potens, MJCM PV 2060; Maledictosuchus riclaensis, MPZ 2001/130a, Parrilla-Bel et al., 2013; Torvoneustes coryphaeus, Young et al., 2013) and in the teleosaurids ?Steneosaurus sp. (SMNS 59558) and Machimosaurus buffetauti (SMNS 91415). This differs from the condition present in Pelagosaurus typus (BSPG 1890 I5), Steneosaurus pictaviensis (LPP.M.37), and Teleosaurus cadomensis (Jouve, 2009) where these structures are incompletely separated. In C. araucanensis the external otic aperture is located posterior to the infratemporal fenestra and opens ventrolaterally. In lateral view the otic aperture is triangular shaped with rounded corners and is completely enclosed within the quadrate. The dorsal margin is overhung by the squamosal, but it does not form part of this margin. The quadrate encloses most of the middle ear cavity.

**Otoccipital.** The otoccipital contacts the supraoccipital dorsally, the parietal and squamosal dorsolaterally, and the quadrate ventrolaterally, and forms the dorsal and lateral margins of the foramen magnum. The foramen magnum is oval, with the major axis mediolaterally oriented. We cannot determine if the otoccipital participates in the dorsal region of the condyle because the sutures are not visible (Fig. 5C). However, in MLP 72-IV-7-4 the otoccipitals participate in the dorsolateral part of the occipital condyle as in most crocodylomorphs. In ventral view it is sutured laterally to the quadrate and medially to the basioccipital (Fig. 5B).

The paroccipital processes are orientated dorsally (Fig. 5C), as in other metriorhynchids (e.g., "M." casamiquelai, MGHF 1-08573; "M." cf. westermanni, MDA 2; Plesiosuchus manselii, Young et al., 2012; Maledictosuchus riclaensis, Parrilla-Bel et al., 2013;

*Torvoneustes coryphaeus*, *Young et al., 2013*). Unlike many crocodylomorphs (e.g., *Protosuchus richardsoni*, UCMP 131827; *Notosuchus terrestris*, MACN-RN 1037; *S. bollensis*, see above; *Caiman yacare*, MACN 15145; *Almadasuchus figarii*, *Pol et al., 2013*) the paroccipital processes are strongly convex on their medial two thirds, while the lateral third is straight. Laterally, the otoccipital contacts the squamosal, through the distal ends of the paroccipital processes. The paroccipital process forms the medial and dorsal borders of the cranioquadrate passage (Fig. 5C).

The ventrolateral flange of the otoccipital is sutured ventrolaterally to the quadrate and ventromedially to the basioccipital. Ventrolateral to the foramen magnum, this region is pierced by several foramina of different diameter. Lateral to the foramen magnum and level with its ventral margin, a small foramen for the passage of CN XII is identified (Fig. 5C). As mentioned above, a single foramen for the exit of CN XII is present in most thalattosuchians. There are three foramina located ventrolateral to the occipital condyle (Fig. 5C). The two lateralmost foramina are identified as the exit of cranial nerves IX, X, XI and associated vessels (e.g., jugular vein). The medialmost foramen, which is ventral to CN IX, X, XI foramina, is not far from the ventral margin of the basioccipital tuberosities and is identified as the enlarged foramen for the internal carotid artery. This foramen is oriented posteriorly in the occipital surface, a condition also observed in other metriorhynchoids (e.g., *Pelagosaurus typus*, BSPG 1890 I5; *Purranisaurus potens*, MJCM PV 2060; *Dakosaurus andiniensis*, MOZ-PV 6146; "*M.*" cf. *westermanni*, MDA 2; *Plesiosuchus manselii*, NHMUK PV R 1089; *Torvoneustes coryphaeus*, *Young et al., 2013*; *Maledictosuchus riclaensis*, MPZ 2001/130a; *Tyrannoneustes lythrodectikos*, *Foffa & Young, 2014*; *Zoneait nargorum*, *Wilberg, 2015a*). In some metriorhynchoid specimens a groove or canal associated with the foramen is ventrally directed (e.g., *Pelagosaurus typus*, BSPG 1890 I5; "*M.*" cf. *westermanni*, MDA 2; *Dakosaurus andiniensis*, MOZ-PV 6146; *Purranisaurus potens*, MJCM PV 2060; and the metriorhynchid specimen LPP.M.23).

**Supraoccipital.** This bone is exposed solely in occipital view. It is flat and subrhomboidal. The supraoccipital is wider (lateromedially) than tall (dorsoventrally) in posterior view (Fig. 5C), as in *Almadasuchus* and crocodyliforms (*Leardi, Pol & Clark, 2017*). Dorsally it contacts the parietal and ventrolaterally the otoccipital. In MLP 72-IV-7-1 the supraoccipital does not contribute to the dorsal margin of the foramen magnum because of the participation of the otoccipitals (Fig. 5C). However, this feature is variable among *C. araucanensis* specimens, as in MLP 72-IV-7-2 and MLP 86-XI-5-7 the supraoccipital reaches the border of the foramen magnum. There is a raised rim in the dorsal region of the supraoccipital–otoccipital suture, that is, interpreted as the occipital tuberosities described by *Brusatte et al. (2016)* for teleosaurids, although these are less pronounced than previously reported. Medial to these tuberosities there are also subtle raised rims, aligned with the vertical walls of the "U"-shaped structure of the parietal (Fig. 5C).

**Basioccipital.** The basioccipital forms the occipital condyle and, as mentioned before, we cannot distinguish if the otoccipital participates in the occipital condyle (Fig. 5C). Ventral to the condyle, in occipital view, the exposed surface is dorsoventrally short

and anteroventrally oriented (Fig. 5C). The basioccipital tuberosities are incompletely preserved because the external surface is eroded; however, they are exposed in posterior and ventral views (Figs. 5B and 5C). In ventral view, the basioccipital tuberosities form a wide "U" with the lateral region in contact with the quadrate. It is triangular and between the tuberosities, it forms the posterior and lateral borders of the median pharyngeal foramen (Fig. 5B).

**Basisphenoid.** In MLP 72-IV-7-1 the basisphenoid is broken and poorly preserved. It is exposed in ventral and lateral view. In ventral view, it contacts the basioccipital posteriorly, laterally the quadrate and anteriorly the pterygoid. The basisphenoid forms the anterior margin of the median pharyngeal foramen (Fig. 5B).

## Endocranial anatomy of *C. araucanensis* (MLP 72-IV-7-1)

**Morphology of the brain endocast.** The cranial endocast of MLP 72-IV-7-1 is complete, from olfactory bulbs to the medulla oblongata and represents approximately 30% of the skull length. It is approximately 141 mm long, from the foramen magnum to the olfactory bulbs, and has a maximum width of 26 mm across the cerebral hemispheres. The brain is elongated, narrow, and relatively straight in lateral view. The dorsal border of the medulla oblongata is almost in the same horizontal plane with the olfactory tract in lateral view, as the midbrain–hindbrain and within hindbrain flexures are not marked (Figs. 7A and 7B). This particular trait has already been noted in previous contributions based on natural brain endocasts (e.g., MLP 76-II-19-1, MOZ-PV 7201; *Herrera, 2015*).

The olfactory tract is long and forms approximately half of the total length of the brain endocast, as in most longirostrine crocodylomorphs (*Pierce, Williams & Benson, 2017*: Figs. 1, 3–4). The olfactory tract widens anteriorly, slightly posterior to the prefrontal pillar, forming the reduced olfactory bulbs. It is worth noting that the pair of large obloid concavities on the ventral surfaces of the frontal that traditionally were interpreted as the olfactory bulbs, actually correspond to the olfactory region of the nasal cavity (see *Herrera, Fernández & Gasparini, 2013*).

In MLP 72-IV-7-1 the cerebral hemispheres are laterally projected and they are noticeably wider (approximately 30%) than the medulla oblongata (Fig. 7C). The same condition was previously observed in the natural endocasts of the same taxon (e.g., MOZ-PV 7201, MOZ-PV 7208, MOZ-PV 7261; *Herrera, 2015*: Fig. 2.2, 2.4), and also noticed in *P. typus* (*Pierce, Williams & Benson, 2017*: Fig. 5A), unlike *S. bollensis* (Fig. 3C) and *S.* cf. *gracilirostris* (*Brusatte et al., 2016*: Figs. 6A–6B).

The anteroposterior elongated pituitary body is located ventral to the midbrain (Figs. 7B and 7F). In lateral view, the pituitary fossa extends from the posterior half of the cerebral hemispheres to pass the posterior margin of the trigeminal foramen (Fig. 7B), as in *S.* cf. *gracilirostris* (*Brusatte et al., 2016*: Fig. 6D). The carotid canals enter the pituitary fossa at its posterior end through two separate foramina and two parallel canals exit anteriorly from the pituitary fossa (Figs. 7B and 7F).

The cranioquadrate passage runs from the middle ear cavity to the cranioquadrate foramen and it conveys the stapedial vein into the middle ear cavity (Fig. 7G), as in

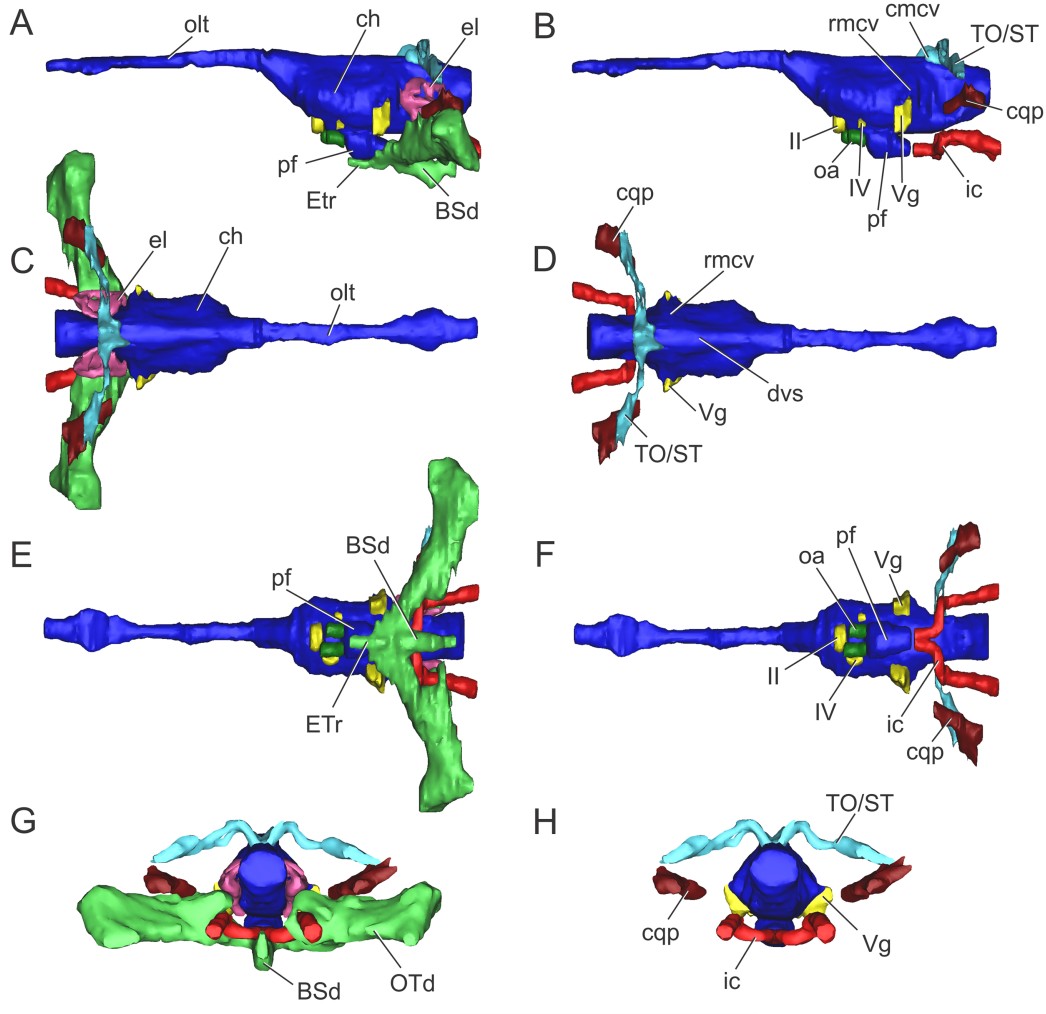

**Figure 7 Endocranial anatomy of *C. araucanensis* (MLP 72-IV-7-1).** In (A and B) lateral; (C and D), dorsal; (E and F), ventral; and (G and H), posterior views. Abbreviations: BSd, basisphenoid diverticulum; ch, cerebral hemisphere; cmcv, caudal middle cerebral vein; cqp, cranioquadrate passage; dvs, dorsal venous sinus; el, endosseous labyrinth of the inner ear; ETr, recessus epitubaricum; ic, internal carotid artery; oa, orbital artery; olt, olfactory tract; OTd, otoccipital diverticulum; pf, pituitary; rmcv, rostral middle cerebral vein; TO/ST, temporo-orbital/stapedial vein; Vg, trigeminal ganglion; II, IV, cranial nerves. Scale bar equals 10 cm.

"*M.*" cf. *westermanni* (*Fernández et al., 2011*) and extant crocodylians (*Porter, Sedlmayr & Witmer, 2016*).

**Vascular elements.** The rostral and caudal middle cerebral veins, the internal carotid artery, and the orbital artery were reconstructed (Fig. 7). The blood vessel fillings distributed throughout the dorsal region of the cerebral hemispheres and associated to the rostral middle cerebral vein identified on the natural casts of *C. araucanensis* and *D.* cf. *andiniensis* (*Herrera, 2015*; *Herrera & Vennari, 2015*) could not be traced in the digital casts based on CT data of MLP 72-IV-7-1.

In lateral view, the rostral middle cerebral vein exits the braincase through the dorsal lobule of the trigeminal foramen, forming a subtle swelling in the endocast of

MLP 72-IV-7-1 (Fig. 7B). This can be related to a low CT resolution, as it is markedly different from the condition of some natural endocasts of *C. araucanensis* where this vein is clearly identifiable (MOZ-PV 7201, MOZ-PV 7261, MLP 73-II-27-3; *Herrera, 2015*: Fig. 2). In dorsal view, and in the hindbrain region, approximately at the level of the endosseous labyrinth, the two branches of the caudal middle cerebral vein exit from the dorsal region of the endocast (Figs. 7B and 7D). These branches are dorsolaterally directed and run dorsally and parallel to the middle ear cavity (Figs. 7A and 7C). The temporo-orbital/stapedial vein passes ventral to the temporo-orbital foramen suggesting that this vein diverges from the main branch and exits through the temporo-orbital foramen. The same feature is also present in "*M.*" cf. *westermanni* (MDA 2), *S.* cf. *gracilirostris* and *Pelagosaurus typus* (*Brusatte et al., 2016*). The temporo-orbital/stapedial vein reaches the middle ear region through the cranioquadrate passage and exits trought the cranioquadrate foramen (Fig. 7H), as it was described for "*M.*" cf. *westermanni* (*Fernández et al., 2011*) and *P. typus* (*Pierce, Williams & Benson, 2017*). Anterior to the root of the caudal middle cerebral veins, the dorsal longitudinal sinus in MLP 72-IV-7-1 is continuous anteriorly as a ridge that overlays the dorsal region of the hind-, mid-, and forebrain (Fig. 7D).

In MLP 72-IV-7-1 the carotid canals were completely reconstructed as these are fully ossified, isolating the internal carotid arteries from the pharyngotympanic sinus (Fig. 7F). This pattern has not been described in any other crocodylomorph. More CT data of metriorhynchids are necessary to confirm if this trait is widespread among Metriorhynchidae or whether it represents an autapomorphy of *C. araucanensis*. In BSPG 1984 I258, *S.* cf. *gracilirostris* (*Brusatte et al., 2016*), and *P. typus* (*Pierce, Williams & Benson, 2017*) the medial portion of the carotid canals passing through the pharyngotympanic sinus cannot be reconstructed. This could be because in these thalattosuchians, as in extant crocodilians, these portions of the carotid canal remain cartilaginous (*Sedlmayr, 2002*). In MLP 72-IV-7-1 the longest portion of the carotid canals runs parallel to the midline of the cranium, from the occipital opening to the level of the cochlear duct. In this region, the carotid canals turn abruptly medially toward the midline of the cranium. The carotid canals then become parallel to the main axis of the skull again before entering at the posterior end of the pituitary fossa through two separate foramina (Fig. 7F). The diameter of the canals at the level of the pituitary fossa is about five mm, slightly thinner than the diameter of the internal carotid foramen (*ca.* six mm).

Clearly separated paired canals exit anteriorly from the pituitary fossa, ventral to the optic nerve (CN II) (Figs. 7B and 7F). The same feature was previously identified in non-metriorhynchid thalattosuchians such as: *S.* cf. *gracilirostris* (*Brusatte et al., 2016*), and *Pelagosaurus typus* (*Pierce, Williams & Benson, 2017*). In MLP 72-IV-7-1 these canals are short (likely because of preservation), with a diameter of approximately five mm, comparable to the diameter of the internal carotid canal. In some natural endocasts (MOZ-PV 7203, MOZ-PV 7205 and MOZ-PV 7261) these paired canals can also be observed in the same region. In recent contributions centered on thalattosuchians, these canals were interpreted as the canals of the orbital artery (*Brusatte et al., 2016*; *Pierce, Williams & Benson, 2017*).

**Nerves.** Only a few cranial nerves were identifiable and reconstructed in MLP 72-IV-7-1. The optic nerve (CN II) originates from the midline, anterodorsal to the pituitary fossa, and exits the braincase through a broad aperture. Cranial nerve IV originates dorsal to the orbital artery (Figs. 7B and 7F). The large trigeminal foramen, located on the lateral wall of the braincase, is a prominent structure that allowed reconstruction of the trigeminal ganglion (Vg), which originates on the lateral surface of the midbrain region (Fig. 7).

**Paratympanic sinus system.** The paratympanic sinus system of *C. araucanensis* resembles that of *S. bollensis*. One of the major differences is the absence of the lateral pharyngeal foramina, a feature shared with most metriorhynchids (e.g., *Purranisaurus potens*, *Metriorhynchus superciliosus*). As in most crocodylomorphs (e.g., *S. bollensis*, *Protosuchus richardsoni*, *Simosuchus shushanensis*, *Notosuchus terrestris*, *Caiman latirostris*), the median pharyngeal foramen is located along the sagittal plane at the suture between the basioccipital and the basisphenoid (Fig. 5B).

The pharyngotympanic sinus system is not expanded posteriorly, thus does not form a basioccipital diverticulum. The absence of this feature is unique among crocodyliforms, even when compared with other thalattosuchians (e.g., *S. bollensis*, see above; *S.* cf. *gracilirostris*, *Brusatte et al., 2016*; *Pelagosaurus typus*, *Pierce, Williams & Benson, 2017*). The median pharyngeal foramen continues anteriorly as a narrow tube, which expands slightly at midlength, pneumatizing the posteroventral region of the basisphenoid (Fig. 7E). From the anterolateral regions of the basisphenoid diverticulum two posterodorsally directed rami, which connect with the main pharyngotympanic sinus, are present. These ramifications are identified as the anterior communicating canals (*sensu Miall, 1878*; or recesus epitubaricus *sensu Dufeau & Witmer, 2015*) and, unlike the morphology present in non-metriorhynchid thalattosuchians, these have little dorsoventral development and are very wide (Fig. 7E). The ventral pneumatization of the floor of the palate is reduced in *C. araucanensis* as it does not invade the pterygoids.

The structure of the main pharyngotympanic sinus of *C. araucanensis* is very similar to that of *S. bollensis*: the middle ear cavity forms a mediolaterally directed tube limited anteriorly by the quadrates and posteriorly by the otoccipitals; accessory pneumatizations from this cavity are very poorly developed (lacking quadrate and infundibular diverticula); and the prootic diverticulum is represented by an anterodorsal swelling on the cavity (Figs. 7A and 7C). The later, as in other thalattosuchians reported to the date (see above), does not form an isolated diverticulum just anterior to the semicircular canals. The main difference identified among *C. araucanensis* and *S. bollensis* is on the relative development of the otoccipital diverticulum, as in the former this diverticulum does not form a laminar ventral expansion (Fig. 7G) as it is observed in *S. bollensis* (Fig. 3G). These laminar ventral expansions of the middle ear cavities (i.e., otoccipital diverticula) are only visible towards the midline, in the posteromedial region of the pharyngotympanic sinus. Another notable difference between the taxa studied in this contribution is the dorsal projection of the otoccipital diverticulum. In *C. araucanensis* this diverticulum is restricted ventral to the foramen magnum. *C. araucanensis* lacks any dorsal enlargement

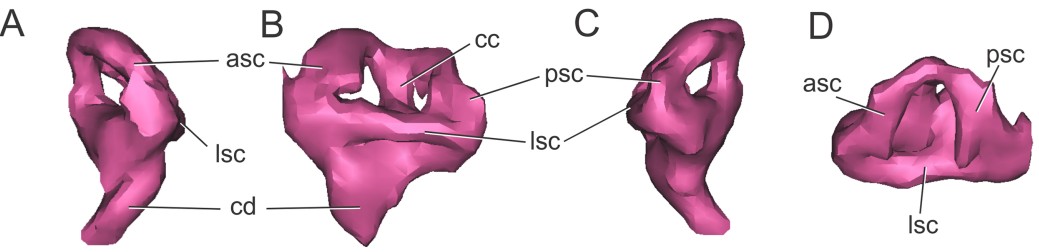

**Figure 8 Left endosseous labyrinth of _C. araucanensis_ (MLP 72-IV-7-1).** In (A) anterior; (B) lateral; (C) posterior; and (D) dorsal views. Abbreviations: asc, anterior semicircular canal; cc, common crus; cd, cochlear duct; lsc, lateral semicircular canal; psc, posterior semicircular canal. Scale bar equals one cm.

or connected cavity (i.e., intertympanic diverticulum) on the posterodorsal region of the paratympanic system, similar to other thalattosuchians (_Brusatte et al., 2016_; _Pierce, Williams & Benson, 2017_).

**Endosseous labyrinth of the inner ear.** The anterior, posterior, and lateral semicircular canals, crus communis, and cochlear duct were reconstructed (Fig. 8). The general morphology of the inner ear is similar to that described in other crocodyliforms. The semicircular canals are aligned in approximately orthogonal planes in three-dimensional space. In MLP 72-IV-7-1, the complete inner ear is approximately 19.5 mm tall and has a maximum width of 18 mm at the level of the semicircular canals (Fig. 8). The cochlear ducts do not extend ventrally in comparison with _S. bollensis_. In MLP 72-IV-7-1 the ventralmost point of the duct is level with the ventral region of the brain (Fig. 7G).

# DISCUSSION

## Comparative braincase and endocranial anatomy

In the sections above we described the braincase and the 3D models of the endocast and other associated structures of two well-preserved thalattosuchians: _S. bollensis_ and _C. araucanensis_. Despite recent phylogenetic studies not using the same taxon sampling (_Pol & Gasparini, 2009_; _Young et al., 2017_), there is a general consensus that the specimens analyzed in this contribution represent members from the two main lineages among thalattosuchians: Teleosauridae and Metriorhynchoidea (see Figs. 9 and 10). This, along with recent contributions centered on the braincase and other associated structures of thalattosuchians (_Fernández et al., 2011_; _Brusatte et al., 2016_; _Pierce, Williams & Benson, 2017_), provided the framework to analyze the main changes in this region in Thalattosuchia. In the following lines we will tackle this issue and try to evaluate the different structures individually.

**Internal carotid foramen and canal.** An enlarged foramen for the internal carotid artery was previously proposed as a synapomorphy of Metriorhynchidae (_Pol & Gasparini, 2009_). However, as mentioned above, this feature is also present in non-metriorhynchid thalattosuchians such as _Teleosaurus cadomensis_, _S._ cf. _gracilirostris_, and _Pelagosaurus typus_ (_Jouve, 2009_; _Brusatte et al., 2016_; _Pierce, Williams & Benson, 2017_), and _S. pictaviensis_ (LPP.M.37) and ?_Steneosaurus_ sp. (SMNS 59558) indicating that an

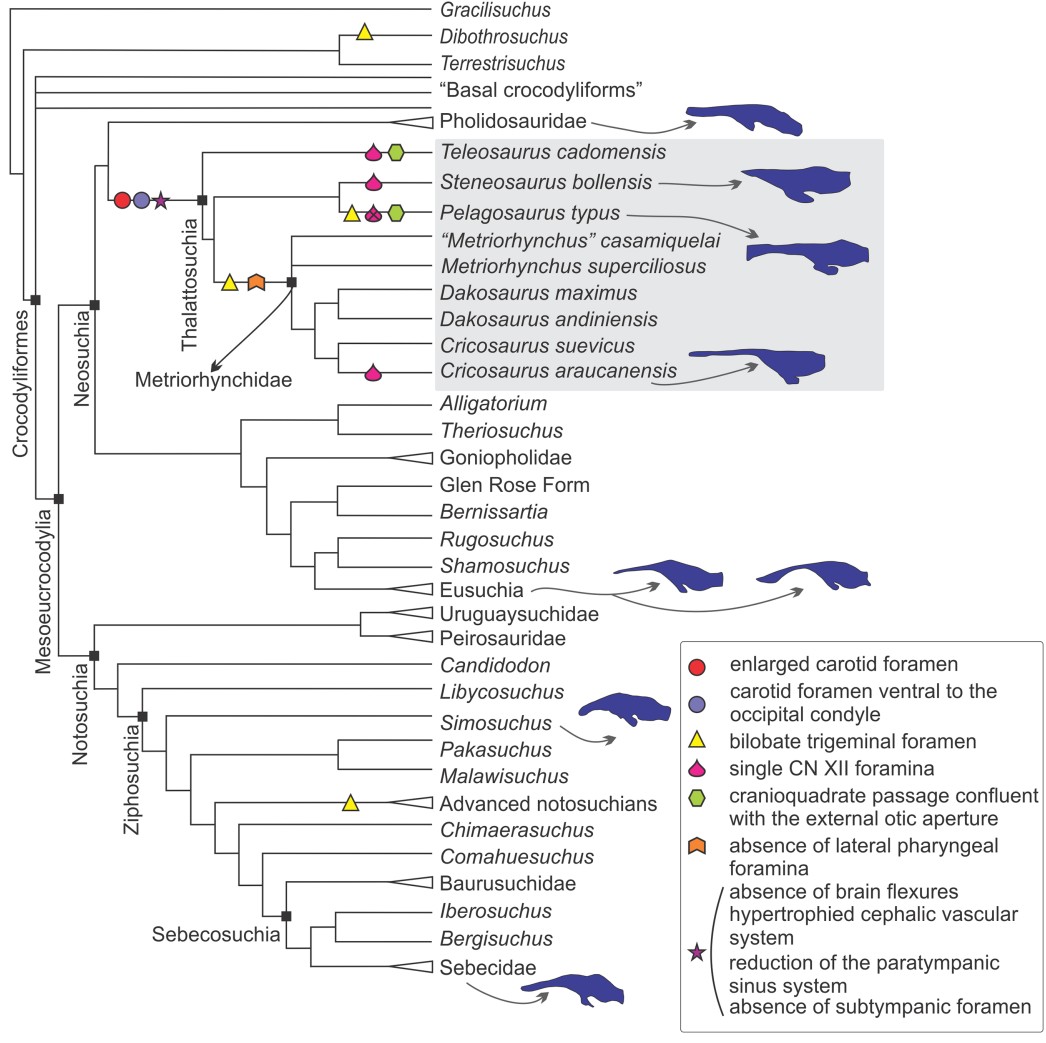

**Figure 9** **Braincase and endocranial features of some crocodylomorphs displayed on a phylogenetic framework (based on** *Pol et al., 2014*). Endocasts were redrawn from the following sources (from top): *Pholidosaurus mereyi* (MB.R.2027), *S. bollensis* (BSPG 1984 I258), *Pelagosaurus typus* (*Dufeau, 2011*), *C. araucanensis* (MLP 72-IV-7-1); *Gavialis gangeticus* (*Bona, Paulina Carabajal & Gasparini, 2017*), *Crocodylus johnstoni* (*Witmer et al., 2008*), *Simosuchus clarki* (*Kley et al., 2010*), *Sebecus icaeorhinus* (*Colbert, 1946a*). Synapomorphies and putative synapomorphies are plotted on the cladogram (see reference chart). An *x* on the symbol represents a reversion on that character. Not to scale.

enlarged foramen for the internal carotid is a feature widely distributed in Thalattosuchia. An exception are the teleosaurids *Machimosaurus buffetauti* (SMNS 91415; *Martin & Vincent, 2013*) and *Peipehsuchus teleorhinus* (IVPP V 10098) where this foramen has the same diameter (or is even smaller) than the one for the exit of the hypoglossal cranial nerve (CN XII).

In metriorhynchids, an enlarged foramen for the internal carotid artery was linked with an enlargement of the carotid canal (*Fernández et al., 2011*). An increased artery diameter implies that there is less blood contacting the vessel wall, thus lowering the friction and the resistance, subsequently increasing the flow. *Herrera, Fernández &*

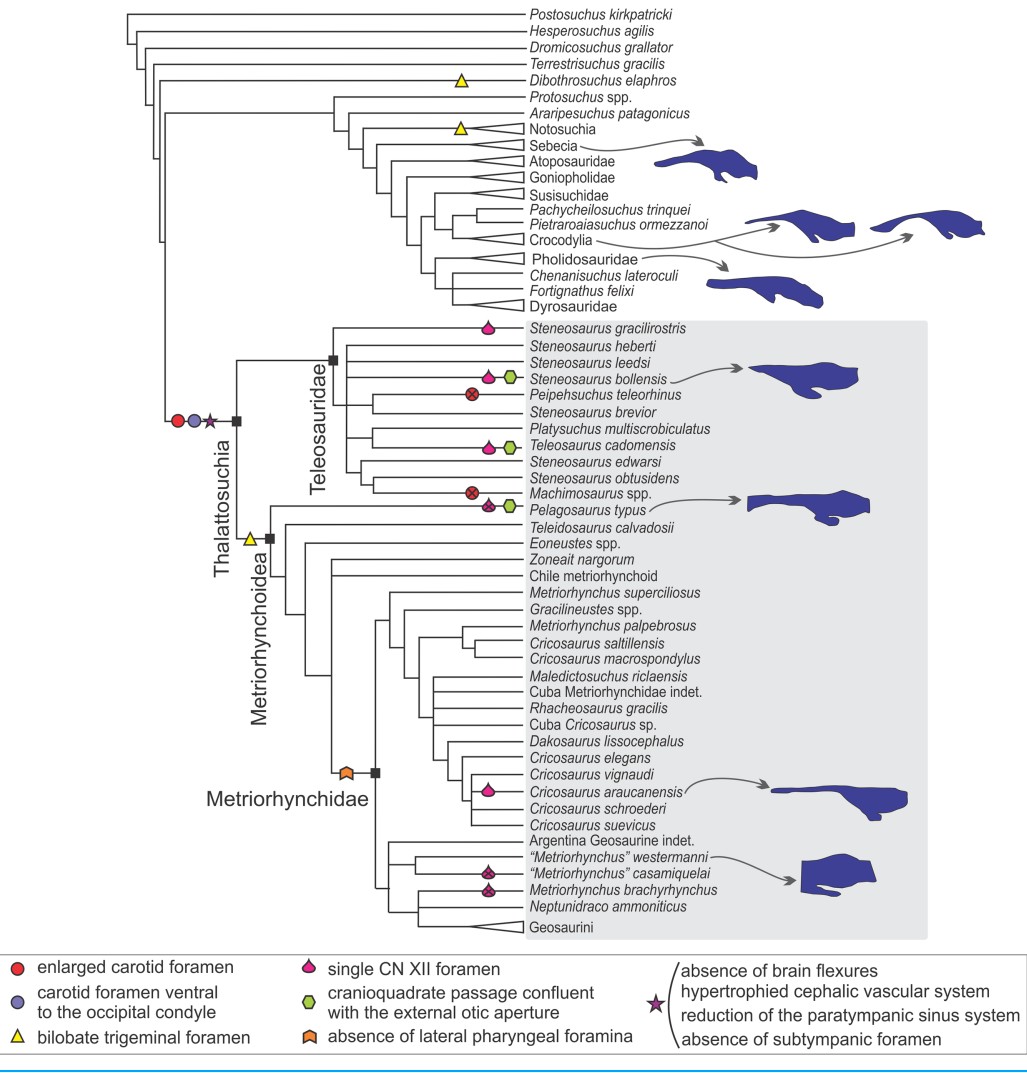

enlarged carotid foramen

carotid foramen ventral to the occipital condyle

bilobate trigeminal foramen

single CN XII foramen

cranioquadrate passage confluent with the external otic aperture

absence of lateral pharyngeal foramina

absence of brain flexures
hypertrophied cephalic vascular system
reduction of the paratympanic sinus system
absence of subtympanic foramen

**Figure 10 Braincase and endocranial features of some crocodylomorphs displayed on a phylogenetic framework (based on *Young et al., 2017*).** Endocasts were redrawn from the following sources (from top): *Sebecus icaeorhinus* (*Colbert, 1946a*); *Gavialis gangeticus* (*Bona, Paulina Carabajal & Gasparini, 2017*), *Crocodylus johnstoni* (*Witmer et al., 2008*), *Pholidosaurus mereyi* (MB.R.2027), *S. bollensis* (BSPG 1984 I258), *Pelagosaurus typus* (*Dufeau, 2011*), *C. araucanensis* (MLP 72-IV-7-1); "*Metriorhynchus*" cf. *westermanni* (MDA 2). Synapomorphies and putative synapomorphies are plotted on the cladogram (see reference chart). An *x* on the symbol represents a reversion on that character. Not to scale.

*Gasparini (2013)* suggested, based on the known case of extant birds (*Gerstberger, 1991*), that in metriorhynchids a high amount of the blood flow through the carotid arteries can be diverted to the glands at maximal salt gland secretion. In this sense, the enlargement of the carotid foramen and canal indicates an increase in blood flow that could be coupled with an increase in blood flow to salt glands at maximal secretion. Recently, *Brusatte et al. (2016)* have used the presence of an enlarged foramen for the internal carotid artery (and also a conspicuous canal) to suggest that large salt glands were also present in the non-metriorhynchid thalattosuchian *S.* cf. *gracilirostris*. More recently, *Wilberg (2015a)* in *Zoneait nargorum* and, *Pierce, Williams & Benson (2017)* in *P. typus*, described an

expansion of the nasal cavity anterior to the orbits and referred this expansion as the osteological correlate of an enlarged salt gland. Again, this claim is coupled with the presence of an enlarged foramen/canal for the internal carotid artery in *P. typus* (*Pierce, Williams & Benson, 2017*). Considering the proposed links mentioned above between the diameter of the carotid foramen and the size of the salt gland, taxa like *Machimosaurus buffetauti* and *Peipehsuchus teleorhinus* would not have enlarged salt glands, if present. Additional to the absence of an enlarged carotid foramen, some specimens of these taxa have been also reported from brackish or continental deposits (*Martin & Vincent, 2013*; *Martin et al., 2016*).

*Fernández & Gasparini (2008)*, following the four stage evolutionary model of osmoregulation strategies of *Dunson & Mazzotti (1989)*, suggested that marine adaptation in thalattosuchians was transitional, that teleosaurids represent the third state, and metriorhynchids the more extreme fourth stage. That is, teleosaurids lived probably in brackish environments and, occasionally in open sea. As the extant *Crocodylus porosus*, teleosaurids likely prevented lethal dehydration by means of small salt-secreting glands of low secretory capacity used in conjunction with selective drinking of only hypo-osmotic fresh water. If so, no conspicuous osteological correlates of salt glands must be expected in teleosaurids. On the contrary, the pelagic life-style of metriorhynchids required well-developed salt-secreting glands, such as those preserved as natural casts in *C. araucanensis* (*Fernández & Gasparini, 2008*).

If enlarged carotid arteries can be used as correlates of an increased blood flow coupled with the blood demand of enlarged salt glands (as proposed by *Brusatte et al., 2016*) then most thalattosuchians would have had enlarged salt glands with a high secretory capacity, and salt glands of low secreting capabilities were restricted to some teleosaurids such as *M. buffetauti* and *P. teleorhinus*. It is worth mentioning that osteological correlates, even in the case of enlarged glands, are not easy to identify. In the case of *C. araucanensis* the identification of these structures was possible as they were preserved as natural casts (see *Fernández & Gasparini, 2008*). It must be noted that the hypothesis about the presence of enlarged salt glands outside Metriorhynchidae is based solely on the indirect evidence of enlarged carotid foramina, and such correlation needs to be validated with fossil specimens displaying enlarged salt glands.

Concerning the position of the foramina for the entrance of the internal carotids, in most thalattosuchians these foramina are situated ventral to the occipital condyle, and lateral to the basioccipital tuberosities, while in most crocodylomorphs these foramina are situated at the level of the occipital condyle (e.g., *Junggarsuchus sloani*, *Clark et al., 2004*: Fig. 2; *Mourasuchus nativus*, *Bona, Degrange & Fernández, 2013*: Fig. 3B; *Caipirasuchus*, *Pol et al., 2014*: Fig. 20B; *Notosuchus terrestris*, *Barrios et al., 2018*: Fig. 22F). Among thalattosuchians there is a modification regarding the general orientation of the carotid canal, which affects in which view the foramina are visible on the skull. In metriorhynchoids, they pierce the otoccipital in a posterior direction, and as a result they are only visible in occipital view (e.g., *Pelagosaurus typus*, *Zoneait nargorum*, *Purranisaurus potens*, *Plesiosuchus manselii*, *Maledictosuchus riclaensis*, *C. araucanensis*, Fig. 5C). This condition is the most widespread among crocodylomorphs, as in most of them (e.g., *Junggarsuchus sloani*, *Mourasuchus*

*nativus, Notosuchus terrestris, Protosuchus richardsoni, Gryposuchus neogaeus, Rhabdognathus aslerensis, Dyrosaurus phosphaticus, Caiman latirostris*), the foramina are visible in occipital view. On the other hand, in the teleosaurids *S. bollensis* (Fig. 1D), *S. pictaviensis, S. leedsi, Machimosaurus buffetauti, P. teleorhinus*, and *Teleosaurus cadomensis* the foramen for the entrance of the carotid artery pierces the otoccipital in a posteroventral, ventrolateral, or even ventral direction. It is worth mentioning that the change in orientation of the internal carotid foramen is only present in the teleosaurids mentioned above, while most of the others exhibit the generalized crocodylomorph condition (e.g., *Steneosaurus* cf. *gracilirostris, Lemmysuchus obtusidens, Steneosaurus edwarsi*). However, it should be considered that in some specimens this feature may be exaggerated by post-mortem deformation because a dorsoventral compression of the skull may cause the internal carotid foramen to be artificially shifted to a more ventral position.

**Trigeminal foramen.** Two different morphologies for the trigeminal foramen have been reported in thalattosuchians: circular (e.g., *S. bollensis, M. buffetauti*, and likely in *T. cadomensis*) (e.g., Fig. 1C) and bilobate, constricted or hour-glass shaped (e.g., *P. typus*, "*M.*" *brachyrhynchus*, "*M.*" cf. *westermanni, C. araucanensis* and *D.* cf. *andiniensis*) (Figs. 6B and 6C). A bilobate trigeminal foramen can be recognized in other non-thalattosuchian crocodylomorphs, for example, *Sphenosuchus* (Walker, 1990), *Dibothrosuchus elaphros* (Wu & Chatterjee, 1993: Fig. 3A), and *Notosuchus terrestris* (Barrios et al., 2018). However, most crocodyliforms have a circular trigeminal foramen (e.g., *Goniopholis stovali*, AMNH 5782; *Caiman latirostris*, MACN 30531; *Protosuchus haughtoni*, Busbey & Gow, 1984; *Rhabdognathus aslerensis*, Brochu et al., 2002; *Simosuchus clarki*, Kley et al., 2010; *Almadasuchus figarii*, Pol et al., 2013).

In sum, a bilobate trigeminal foramen appears to be restricted to metriorhynchoids within Thalattosuchia, whereas teleosaurids retain the plesiomorphic condition of a circular trigeminal foramen (Figs. 9 and 10). Although preliminary, as this feature needs to be tested thoroughly in a phylogenetic context, occurrences of bilobate trigeminal foramina seem to be isolated outside Metriorhynchoidea, contrasting with the widespread condition of a circular one. This particular morphology has been associated to the separation of the maxillary (CN $V_2$) and mandibular (CN $V_3$) branches of the CN V (Barrios et al., 2018).

**Cranial nerves IX–XI and XII.** The presence of a metotic foramen with a unique or double opening has been recognized within Thalattosuchia. We interpreted that *S. bollensis* (Fig. 1D), *Pelagosaurus typus, Purranisaurus potens*, and *Dakosaurus* cf. *andiniensis* have a single opening. In the holotype of "*Metriorhynchus*" *casamiquelai* (MGHF 1-08573) there is also one opening for these cranial nerves, while Soto-Acuña, Otero & Rubilar-Rogers (2012: Fig. 2) identified two foramina for the exit of the CN IX-XI in other specimen referred to "*Metriorhynchus*" *casamiquelai*. In contrast, in *C. araucanensis* (Fig. 5C), the metriorhynchid specimen from Mörnsheim Formation (BSPG 1973 I195), "*M.*" *westermanni*, "*M.*" cf. *westermanni*, "*M.*" *brachyrhynchus* (LPP.M 22), and *Teleosaurus cadomensis* there is a double opening for the exit of cranial nerves IX, X, and XI.

Related to the exit for CN XII, most thalattosuchians have only one foramen, except for *Pelagosaurus typus*, "*M.*" *brachyrhynchus* (LPP.M 22), and likely "*M.*" *casamiquelai*.

The presence of a single foramen is shared with other crocodylomorphs (e.g., *Gryposuchus neogaeus*, some *Rhabdognathus* species). Although it is a different condition in comparison with most crocodylomorphs (e.g., *Protosuchus*, *Sphenosuchus*, *Almadasuchus*, *Notosuchus terrestris*, *Dibothrosuchus elaphros*) that have two foramina.

It must be considered the possibility that in some cases where more than one foramina is interpreted as the exit for CN IX–XI, the medialmost could corresponds to a second opening for CN XII. This uncertainty can be resolved with complete reconstructions of the passage of the nerves, based on CT data, but unfortunately in the specimens that we are describing this is not possible because the resolution of CT data does not allow tracing of these passages. Within the studied thalattosuchians, there is variability in the number, relative size, or location of the cranial nerve foramina on the posterior aspect of the skull (see also *Brusatte et al., 2016*).

**Separation of the cranioquadrate canal from the external otic aperture.** The cranioquadrate canal is a structure present in hallopodid crocodylomorphs, mesoeucrocodylians and derived crocodyliforms (i.e., *Fruitachampsa*, *Gobiosuchus*, *Hsiosuchus*) (*Clark, 1994*; *Leardi, Pol & Clark, 2017*). The cranioquadrate canal connects the middle ear space with the posterior aspect of the skull, which additionally serves to transmit the stapedial artery and vein (*Iordansky, 1973*; *Porter, Sedlmayr & Witmer, 2016*). In thalattosuchians the cranioquadrate foramen is placed more laterally than in most crocodyliforms, a condition reported as synapomorphy for the group (*Clark, 1994*).

In this contribution we report a further derived condition for the cranioquadrate passage present in all metriorhynchids and some telesaurids, in which the cranioquadrate foramen is completely separated from the external otic aperture by a thin bony lamina. This condition is present in the metriorhynchids *C. araucanensis* (Fig. 5C), *Maledictosuchus riclaensis*, *Torvoneustes coryphaeus*, *Purranisaurus potens*, "*Metriorhynchus*" *casamiquelai*, and in the teleosaurids *Machimosaurus buffetauti* and ?*Steneosaurus* sp. (SMNS 59558). Aditionally, *Jouve (2009)* mentioned that these structures are also completely separated in *Mystriosaurus* cf. *bollensis*, *Teleidosaurus*, and *Enaliosuchus*. In contrast, based on first hand examinations of the specimens, we found that the cranioquadrate canal is incompletely separated from the external otic recess in *Pelagosaurus typus* (BSPG 1890 I5, NHMUK PV OR 32599) and *Steneosaurus pictaviensis* (LPP.M.37). According to *Jouve (2009)* the same feature ocurrs in *Teleosaurus cadomensis*, *Steneosaurus larteti*, *Pelagosaurus typus*, and *S. bollensis*. The distribution of this character within non-metriorhynchid thalattosuchians appears to be a highly homoplastic trait. However, it should be considered that the complete separation of the cranioquadrate canal and the external otic recess in some non-metriorhynchid thalattosuchians could be cartilaginous. This is based on our observations of the specimen *Pelagosaurus typus* (BSPG 1890 I5) where the separation of both structures is incomplete, but the bony laminae almost contact each other. Yet, the separation is very difficult to evaluate in internal structures in fossil taxa, as the osteological marks left by the cartilage can be very subtle (*Holliday & Witmer, 2008*).

**Absence of flexures.** Most extant and extinct crocodylomorphs have, at least, a well-marked mid-hindbrain flexure, giving the encephalon a curved profile in lateral

view (e.g., *Pholidosaurus meyeri*, MB.R.2027; *Macelognathus*, *Leardi, Pol & Clark, 2017*; *Sebecus icaeorhinus*, *Colbert, 1946a*: pl. 14A; *Crocodylus johnstoni*, *Witmer et al., 2008*: Fig. 6.3; *Simosuchus clarki*, *Kley et al., 2010*: Fig. 32C; *Gavialis gangeticus*, *Bona, Paulina Carabajal & Gasparini, 2017*: Fig. 7). In this sense, the presence of a straight brain (i.e., absence of flexures between fore-midbrain, and mid-hindbrain), appears to be characteristic for Thalattosuchia (*Herrera, 2015*; *Herrera & Vennari, 2015*; *Brusatte et al., 2016*; *Pierce, Williams & Benson, 2017*).

The tubular brain of thalattosuchians can be coupled with two morphological traits present in their skulls. Most thalattosuchians have a very long and tubular snout which does not have an abrupt transition between its dorsal edge and the anterior end of the skull roof (i.e., the anterior end of the frontal). This feature is particularly marked in *C. araucanensis* (*Gasparini & Dellapé, 1976*: Lam, 1–2), where the anterodorsal part of the snout is almost straight with the skull roof. This particular transition between the skull roof and the snout has been previously noted in other non-thalattosuchian crocodylomorphs like *Sebecus icaeorhinus* and baurusuchids (*Colbert, 1946a*; *Carvalho, Campos & Nobre, 2005*). Also, *Sebecus* has a fore-midbrain flexure with an angle comparable to the one observed in thalattosuchians (see *Pierce, Williams & Benson, 2017*). On the other hand, the mid-hindbrain flexure could be hidden by the development of the large dorsal venous sinus on the posterodorsal region of the encephalon, that overlies this region and obscures the real nature of the flexure. However, these interpretations need to be thoroughly tested as other taxa like *Simosuchus* have acute angles between parts of its encephalon attaining values similar to the ones present in thalattosuchians (*Pierce, Williams & Benson, 2017*). It is important to note that, to the present day, CT data and 3D endocast models for crocodylomorphs are very few and more data could easily change the interpretations presented in here.

**Cephalic vascularization.** The cephalic vascular system in Thalattosuchia is characterized by a well-developed caudal middle cerebral vein/stapedial vein/temporo-orbital vein, internal carotid artery, and orbital artery (see also *Fernández et al., 2011*; *Herrera, Fernández & Gasparini, 2013*; *Herrera, 2015*; *Herrera & Vennari, 2015*; *Herrera, 2016*; *Brusatte et al., 2016*; *Pierce, Williams & Benson, 2017*). Additionally, descriptions of natural brain endocasts of metriorhynchids (*Herrera, 2015*; *Herrera & Vennari, 2015*) showed that several blood vessels cover the dorsal region of the cerebral hemispheres, and are associated with the rostral middle cerebral vein. This enlarged vascular system indicates an important blood supply to the head of thalattosuchians.

A plausible functional explanation about the enlarged cephalic vascular system in Thalattosuchia is that it can play a role in cephalic physiological thermoregulation. The orbital artery supplies the posterior aspect of the orbit, whereas the venous drainage of the orbit is mainly via the temporo-orbital veins. Extant crocodilians have the orbital region well vascularized and arteries and veins form a plexus within the orbit, potentially allowing heat exchange (*Porter, Sedlmayr & Witmer, 2016*). In *C. araucanensis* and *S.* cf. *gracilirostris*, the orbital artery is approximately the same diameter as the internal carotid artery. In both taxa and also in "*Metriorhynchus*" cf. *westermanni*, *Dakosaurus* cf.

*andiniensis*, and *Pelagosaurus typus* the temporo-orbital vein is also well developed. This suggests that these blood vessels were capable of transmitting a large volume of blood to the orbit. Blood shunts or countercurrent heat exchangers have been described for several species of reptiles as responsible for regional temperature differences (*Heath, 1966*; *Crawford, Palomeque & Barber, 1977*). The blood vessels infillings that cover the cerebral hemispheres and derive from the rostral middle cerebral vein described in metriorhynchids (*Herrera, 2015*; *Herrera & Vennari, 2015*) could be acted as heat exchangers. As mentioned by *Porter, Sedlmayr & Witmer (2016)*, the anatomical and physiological roles that blood vessels play in extant crocodylian thermoregulation need further investigation. Other funtional alternative is that the enlarged cephalic vascular system could be related with a mitigation of the impact of hydrostatic pressures on organs and neural tissues. However, the exploration of this alternative require further exploration and it is beyond the scope of the present contribution.

**Lateral pharyngeal foramen (pharyngotympanic tube).** The pharyngotympanic tubes connect the middle ear cavity with the pharynx and are part of the mechanism used in several amniotes to equalize pressures of the middle ear and the external environment (*Dufeau & Witmer, 2015*).

In metriorhynchids the lateral pharyngeal foramina are closed as the basisphenoid contacts the otooccipital and the basioccipital along its posterolateral edge (Fig. 5B). On the other hand, teleosaurids (e.g., *S. bollensis*, BSPG 1984 I258, Fig. 1B; *S. pictaviensis*, LPP. M.37; *Lemmysuchus obtusidens*, LPP.M.21; *Teleosaurus cadomensis*, *Jouve, 2009*) and the basal metriorhynchoid *Pelagosaurus typus* (BSPG 1890 I5) bear lateral Eustachian foramina, as most crocodyliforms (e.g., *Protosuchus haughtoni*, *Notosuchus terrestris*, *Rhabdognathus aslerensis*, *Caiman latirostris*).

Within Thalattosuchia the closure of the lateral pharyngeal foramina appears to be restricted to metriorhynchids, while non-metriorhynchid thalattosuchians retain the plesiomorphic condition of bearing lateral pharyngeal foramina. Besides the closure of the lateral pharyngeal foramina, additional variation can be reported in thalattosuchians. In teleosaurid taxa, restricted to the genus *Steneosaurus* (e.g., *S. bollensis*, BSPG 1984 I258; *S. pictaviensis*, LPP.M.37), the median pharyngeal foramen is smaller than lateral ones, while *Peipehsuchus teleorhinus* (IVPP V 10098) retains the typical crocodyliform condition where the medial foramen is the largest of the pharyngeal foramina.

The absence of the lateral pharyngeal foramina in metriorhynchids implies that the communication between the middle ear cavity with the pharynx is reduced to the median pharyngeal foramen. Further studies are needed to understand if this modification has a physiological/adaptive significance.

**Reduction of the paratympanic sinus system.** According to *Dufeau (2011)* and *Dufeau & Witmer (2015)* in most mesoeucrocodylian crocodyliforms (*Sebecus icaeorhinus*; *Hamadasuchus rebouli*; and extant crocodiles like *Alligator mississippiensis*), diverticular expansions are extensive whereas in longirostrine taxa such as *Gavialis*, *Tomistoma*, *Rhabdognathus aslerensis* and *Pelagosaurus* there is a constraint on diverticular pneumatization of the anterior portion of the braincase.

The constraint on diverticular pneumatization is more evident in Thalattosuchia, in comparison to extant and extinct crocodiles, given by the absence of the infundibular, quadrate, intertympanic diverticula, likely the prootic diverticulum and also a restricted pneumatization of the otoccipital (see also *Brusatte et al., 2016*; *Pierce, Williams & Benson, 2017*). This reduction is more developed in *C. araucanensis* than in *S. bollensis*. The former also lacks a basioccipital diverticulum and the otoccipital diverticulum has no dorsal projection (in comparison to *S. bollensis*). Future work should confirm if this condition is restricted to *C. araucanensis* or if it is more widely distributed among metriorhynchids.

The development of pneumatizations anterior to the tympanic crest is not so well documented in non-crocodyliform crocodylomorphs, although pneumatic anterior foramina have been reported in several taxa (e.g., *Terrestrisuchus*, *Dibothrosuchus*, *Junggarsuchus*, *Almadasuchus*, *Macelognathus*), and in some of them the connection with the middle ear cavity was confirmed (*Leardi, Pol & Clark, 2017*). As it was observed with the peculiar shape of the otic aperture, the lack of a subtympanic foramen (*sensu Montefeltro, Andrade & Larsson, 2016*) and an associated infundibular diverticulum is common to most thalattosuchians (e.g., *S. bollensis*, *T. cadomensis*, *P. typus*, *C. araucanensis*).

The lack of pneumatic features on the supraoccipital (intertympanic sinus) has been proposed as evidence for a non-crocodyliform position for Thalattosuchia in previous contributions (*Clark, 1986*, *1994*; *Wilberg, 2015b*). However, the lack of a prootic pneumatization in thalattosuchians, a feature widely distributed among derived crocodylomorphs (see above), has gone unnoticed in recent analysis of the paratympanic pneumaticity of thalattosuchians (*Wilberg, 2015b*; *Brusatte et al., 2016*; *Pierce, Williams & Benson, 2017*). The lack of dorsal communication via an intertympanic sinus could be representing a generalized loss of paratympanic pneumaticity in thalattosuchians, an idea supported by the condition seen in other regions of the braincase (e.g., otoccipital). Thus, more information is needed to understand the thalattosuchian changes in pneumaticity to support the claim that the absence of the intertympanic sinus is due to a non-crocodyliform position within the crocodylomorph phylogeny or if it represents a regression.

The posterior pneumatization of the otic capsule is present and well-developed in derived non-crocodyliform crocodylomorphs and crocodyliforms (*Pol et al., 2013*; *Leardi, Pol & Clark, 2017*), where the otoccipital diverticulum is dorsally extended, exceeding the dorsal border of the foramen magnum. This contrasts with the thalattosuchian condition, in which the diverticulum is mostly restricted to the ventral part of the otoccipital.

Pneumatic diverticula associated with the middle ear cavity of crocodylomorphs have been described and analyzed several times (*Colbert, 1946b*; *Tarsitano, 1985*; *Dufeau & Witmer, 2015*). However, the functional interpretations related to this particular crocodylomorph specialization remain elusive. Given that interaural time differences have been reported for crocodylians (*Carr et al., 2009*), *Bierman et al. (2014)* hypothesized that the pneumatic diverticula coupled both ears internally, allowing the crocodylian ear to act as a pressure difference receiver organ and, thus, permitting directional hearing.

A second hypothesis was proposed by *Dufeau & Witmer (2015)*, in which the development of the paratympanic pneumatic diverticula increases the auditory sensitivity. In particular, *Dufeau & Witmer (2015)* found that the resonant frequencies calculated on the subtympanic foramen coincided with the greatest intensity of juvenile vocalizations of *Alligator mississippiensis*, while these distress calls where among the lowest threshold of cochlear sensitivity in adults. Thus, it was associated to have a function in increasing the auditory sensitivity from adults toward their offspring. Due to the recent increasing amount of available CT data of thalattosuchians we can conclude that the group as a whole lacks the intertympanic diverticulum, and has a reduced dorsal pneumatization associated with the otic capsule (see above). However, internal connections between both middle ears are still retained through the ventral part of the pharyngotympanic and median pharyngeal systems, thus allowing internal ear coupling and the associated sound localization. As a result, we can infer a reduced response in directional hearing in thalattosuchians and a decrease in low frequency sensitivity (*Bierman et al., 2014*), due to the loss of the dorsal paratympanic pneumatization. On the other hand, the increased sensitivity hypothesis through the subtympanic foramen can be discarded for thalattosuchians. As it was discussed in this contribution, thalattosuchians lack pneumatic foramina and associated pneumatization in the quadrate.

## Braincase and endocranial anatomy evolution of thalattosuchians

In order to trace morphological transformations and major changes in the braincase and endocranial anatomy along crocodylomorph evolution, the distribution of the anatomical features that we discussed above were mapped in two phylogenetic hypotheses for the clade: Thalattosuchia as derived neosuchians, nested with pholidosaurs/dyrosaurids forming a "longirostrine clade" (Fig. 9; *Clark, 1994*; *Pol & Gasparini, 2009*; *Leardi, Pol & Clark, 2017*); or Thalattosuchia as the sister group to Crocodyliformes (Fig. 10; *Wilberg, 2015b*; *Young et al., 2017*). This approach is due to the unresolved phylogenetic affinities of Thalattosuchia.

Under both phylogenetic hypotheses the enlarged foramen for the internal carotid, the carotid foramen ventral to the occipital condyle, an orbital process of the quadrate free of bony attachment, a laterosphenoid-prootic suture forming a pronounced ridge, the absence of a subtympanic foramen, the absence of brain flexures, the hypertrophied cephalic vascular system (i.e., the enlargement of the carotids and orbital arteries as well as temporo-orbital veins), and the reduction of the paratympanic sinus system are putative synapomorphic features of Thalattosuchia (Figs. 9 and 10). Thalattosuchians are very diverse and some of these features exhibit reversions, showing instances of homoplasies (e.g., double foramen for the exit of CN XII in some thalattosuchians and the absence of an enlarged carotid foramen in *Machimosaurus* and *Peipehsuchus*). However, some of the traits discussed are unknown in many crocodylomorphs, either due to the lack of well-preserved braincases, incomplete descriptions in the literature of this area, or the lack of internal anatomical data (e.g., CT data).

Although a general pattern of braincase configuration is evident in Thalattosuchia, other morphological traits characterize less inclusive clades. Such is the case with the

shape of the trigeminal foramen. In the topology of *Pol et al. (2014)* (Fig. 9) the bilobate trigeminal foramen characterizes Metriorhynchidae, although a bilobate trigeminal foramen is also present in the non-metriorhynchid thalattosuchian *Pelagosaurus typus*. On the other hand, based on the topology of *Young et al. (2017)* the bilobate trigeminal foramen is a putative synapomorphy of Metriorhynchoidea. In both phylogenetic hypotheses the loss of the lateral Eustachian foramina is a putative synapomorphy of Metriorhynchidae (Figs. 9 and 10). This feature could be coupled with the general reduction evidenced in the paratympanic sinus system (absence of basioccipital and pterygoid diverticula) of *C. araucanensis*. However, in order to evaluate if this condition is extended among other members of Metriorhynchidae, futher descriptions of the metriorhynchid paratympanic sinus system are required.

## CONCLUSIONS

The braincase and endocranial morphology of the teleosaurid *S. bollensis* and the metriorhynchid *C. araucanensis* are described in this contribution. The descriptions of two members of different clades of Thalattosuchia allowed us to evaluate and contrast the main features of the braincase and endocranial anatomy with other crocodylomorphs. The main traits that characterize Thalattosuchia from other crocodylomorphs are: enlarged foramen for the internal carotid artery, the carotid foramen ventral to the occipital condyle, a single CN XII foramen, the absence of brain flexures, the well-developed cephalic vascular system, the absence of a subtympanic foramen, and the reduction of the paratympanic sinus system. Some of these features (enlarged foramen for the internal carotid artery, the absence of brain flexures, the hypertrophied cephalic vascular system), were previously suggested as exclusively present in Metriorhynchidae, and associated to the pelagic lifestyle of this lineage; however, recent studies revealed that they were already established at the base of Thalattosuchia.

From the paleobiological perspective, these changes indubitably had consequences for the biology of these animals. We suggest that the well-developed vascular system was not only related to the secretory function of salt glands, but also it played a role in the cephalic physiological thermoregulation. Also, the reduction of the paratympanic pneumatization is related to a reduced response in directional hearing in thalattosuchians and a decrease in low frequency sensitivity. However, these interpretations should be tested in the light of new information about extant and extinct archosaurs.

As it was mentioned above, the main modifications on the braincase and endocranial anatomy appear to be present even in the basalmost members of the clade. These findings do not support an adaptive gap between fully pelagic forms (metriorhynchids) and semiaquatic ones (teleosaurids), implying that these features were already present in the lineage and might have been exapted later on their evolutionary history.

We recognized differences within Thalattosuchia that previously have not received much attention or even were overlooked (e.g., circular/bilobate trigeminal foramen, single/double CN XII foramen, separation of the cranioquadrate canal from the external otic aperture, absence/presence of lateral pharyngeal foramen). The new information

on the braincase and endocranial morphology of thalattosuchians adds anatomical information that has potential use in taxonomy, phylogeny, and paleobiology.

The functional significance of these traits is still unclear. Extending the sampling to other thalattosuchian taxa will help to test the timing of acquisition and distribution of these morphological modifications among the whole lineage. Also, comparison with extant marine tetrapods (including physiological information) will be crucial to understand if some (and/or which) of the morphological peculiarities of thalattosuchian braincases are products of directional natural selection resulting in full adaptation to a nektonic life style.

## INSTITUTIONAL ABBREVIATIONS

**AMNH**  American Museum of Natural History (Fossil Reptiles), New York, USA

**BSPG**  Bayerische Staatssammlung für Paläontologie und Geologie, Munich, Germany

**GPIT**  Paläontologische Sammlung der Eberhard Karls Universität Tübingen, Tübingen, Germany

**IVPP**  Institute of Vertebrate Paleontology and Paleoanthropology, Chinese Academy of Sciences, Beijing, China

**LPP**  Institut de paléoprimatologie, paléontologie, humaine; évolution et paléoenvironnements Université de Poitiers, Poitiers, France

**MACN**  Museo Argentino de Ciencias Naturales "Bernardino Rivadavia," Buenos Aires, Argentina

**MB.R.**  Museum für Naturkunde Humboldt-Universtät, Berlin, Germany

**MDA**  Museo del Desierto de Atacama, Antofagasta, Chile

**MGHF**  Museo Geológico H. Fuenzalida, Universidad Católica del Norte, Antofagasta, Chile

**MJCM**  Museo de Ciencias Naturales y Antropológicas "Juan Cornelio Moyano," Mendoza, Argentina

**MLP**  Museo de La Plata, La Plata, Argentina

**MOZ**  Museo Provincial de Ciencias Naturales "Prof. Dr. Juan A. Olsacher," Zapala, Neuquén, Argentina

**MPZ**  Museo Paleontológico de la Universidad de Zaragoza, Zaragoza, Spain

**NHMUK**  Natural History Museum, London, UK

**SMNS**  Staatliches Museum für Naturkunde, Stuttgart, Germany

**UCMP**  University of California Museum of Paleontology, Berkeley, USA.

## ACKNOWLEDGEMENTS

We thank C. Mehling and M. Norell (AMNH), O. Rauhut (BSPG), W. Joyce, P. Havrlik and M. Aiglstorfer (GPIT), F. Zheng and X. Xu (IVPP), P. Vignaud and G. Garcia (LPP), A. Kramarz (MACN), T. Schossleitner and D. Schwarz (MB.R.), A. Garrido and B. Bollini (MOZ), J.I. Canudo (MPZ), L. Steel (NHMUK), R. Schoch (SMNS), and P. Holroyd (UCMP) for provided access to specimens under their care and valuable

help during collection visits. We thank G. Rößner (BSPG) for providing assistance with CT scanning. We would like thank F. Degrange (CICTERRA) and M. Bronzati (FFCLRP-USP) for their technical support. We are also grateful to S. Walsh, G. Sobral and E. Wilberg for constructive comments and valuable insights, and Fabien Knoll for handling the manuscript. YH deeply thanks O. Rauhut (BSPG) for his support during her stay in Munich. This is JML's R-255 contribution to the Instituto de Estudios Andinos Don Pablo Groeber.

### Funding

This research has been supported by the following grants: a Humboldt Research Fellowship for Postdoctoral Researchers from the Alexander von Humboldt Foundation and a Scholarship Programme for Young Professors and Researchers from Latin American Universities from Coimbra Group to Yanina Herrera; Agencia Nacional de Promoción Científica y Tecnológica (PICTs 2016-0267, 2016-1039); and Programa de Incentivos Universidad Nacional de La Plata N775 (Argentina). There was no additional external funding received for this study. The funders had no role in study design, data collection and analysis, decision to publish, or preparation of the manuscript.

### Grant Disclosures

The following grant information was disclosed by the authors:
Humboldt Research Fellowship for Postdoctoral Researchers from the Alexander von Humboldt Foundation.
Coimbra Group Scholarship Programme for Young Professors and Researchers from Latin American Universities.
Agencia Nacional de Promoción Científica y Tecnológica: PICTs 2016-0267, 2016-1039.
Programa de Incentivos Universidad Nacional de La Plata N775 (Argentina).

### Competing Interests

The authors declare that they have no competing interests.

### Author Contributions

- Yanina Herrera conceived and designed the experiments, performed the experiments, analyzed the data, contributed reagents/materials/analysis tools, prepared figures and/or tables, authored or reviewed drafts of the paper, approved the final draft.
- Juan Martín Leardi conceived and designed the experiments, performed the experiments, analyzed the data, contributed reagents/materials/analysis tools, prepared figures and/or tables, authored or reviewed drafts of the paper, approved the final draft.
- Marta S. Fernández conceived and designed the experiments, performed the experiments, analyzed the data, authored or reviewed drafts of the paper, approved the final draft.

## Data Availability

MorphoSource: https://www.morphosource.org/Detail/ProjectDetail/Show/project_id/528.

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
