# Peer review of "Braincase and endocranial anatomy of two thalattosuchian crocodylomorphs and their relevance in understanding their adaptations to the marine environment"

_PeerJ, doi:10.7717/peerj.5686_

## Round 0.1 · original submission · Minor Revisions

Dear Yanina,

I have now received three reviews of your article submitted to PeerJ. The reviewers are overall positive, but recommend a number of improvements that should be addressed before re-submission.
I was unable to download the CT data folders from MorphoBank. Please make sure these are available to the reviewers in the second round of reviews.

Please, together with your unmarked revised manuscript, provide a marked-up copy as well as a document explaining how you have addressed each of the points raised by the reviewers.

Best regards,
Fabien

·

Basic reporting

The English is very good on the whole - I have highlighted minor issues in my annotated version of the MS PDF.

Ideally, the original tomograph image stack should be made publicly available for independent verification of the authors' results, but mesh models of the segmentations should be included as electronic supplementary data in STL, PLY or 3D PDF format.

Experimental design

No comment.

Validity of the findings

No comment.

Additional comments

Review: Herrera, Leardi and Fernández.

This manuscript is on the whole well written and easy to follow, with good descriptive sections on both the osteology and neuroanatomy. There are a few issues with language (agreement for plurals and verbs etc.), but these are very minor and I have annotated the PDF version of the MS to indicate where changes can or should be made. The descriptive detail alone makes this MS a valuable contribution to the field, and I have no real reservations recommending it for publication with minor corrections

I have a few comments about the content that would need to be dealt with, mostly by adding a little extra discussion or comment.

1. MLP 72-IV-7-1 was scanned at low resolution on a medical scanner. The authors rightly mention possible limitations in analysis and interpretation that this may have caused at various points in the text. I think the resolution is probably sufficient given the size of the specimen, but this issue should probably be discussed in the text more thoroughly. The voxel dimensions must be given (lns 124-5) for consistency with scan parameters given for BSPG 1984 1258.

2. A statement about likely ontogenetic stage is needed for both specimens, as this affects the fidelity of the brain cavity endocast for the brain itself (lns 298-318).

3. Endocast figures – is there any point in including ‘en’ as an annotation? If it is used it would need to be ‘brain cavity endocast’ because the rest of the segmentations are endocasts too.

4. The shape of the pituitary fossa should not be assumed to be the shape of the pituitary itself – the shape of the fossa in BSPG 1984 1258 is clearly affected by the anastomosis of the carotid rami, and the rostral projection of the arterial pathway.

5. The relative length of the endosseous cochlear canal appears to be markedly different in each taxon, but there is no real discussion of what this might mean with respect to hearing sensitivity thresholds (e.g., Walsh et al. 2009; doi:10.1098/rspb.2008.1390). Could the shorter length in MLP 72-IV-7-1 be an artefact of the lower resolution scan? This could be discussed. There seems to be surprisingly little discussion of the labyrinth in either specimen despite the emphasis that previous research has placed on the structure with respect to adaptation during evolutionary transitions out of and into water. This really could be expanded on. See Benson et al. (2017; doi: 10.1111/joa.12726) for discussion of SSC canal shape and size in birds for some discussion of limitations.

6. The olfactory tract projects from the olfactory bulbs to centres in the rest of the telencephalon – in most sauropsids there is probably no trace of the tract on endocasts because the connection is internal, and all that can be seen is the olfactory bulb and cerebral hemispheres. There may be no obvious demarcation on the endocast between the two even in birds, and the endocasts of these particular crocodylomorphs are especially difficult to interpret. I have to admit that I am uncomfortable with the idea of the telencephalon being stretched in such a way that the olfactory bulbs end up separated so far away from the cerebral hemispheres because the rostrum lengthens – all of the brain should be enclosed by the skull roof. In fact, evolutionary expansion of brain regions tends to create a problem because the interconnecting ‘wiring’ has to become longer. Lengthening CNI as the olfactory epithelium moves rostrally during snout lengthens makes more sense. Are you sure this is the olfactory tract on both endocasts?

In Steneosaurus (Figure 3 A and C), there appears to be a slight bulge rostral of the taper of the telencephalon. If this is the case, the structure labelled as the olfactory tract would have to be a joined ramus for CNI. However, CNI would be rather thick if this is the case. Likewise, if the widening marked ‘olt’ in Cricosaurus (Figure 7 A and B) is actually the olfactory region of the nasal cavity, the ‘olt’ has to be CNI. This then begs the question of what projects rostrally from the nasal cavity? Either way, the labelling for the figures would need either a slight tweak or a complete review.

7. Discussion: As the discussion stands, it is ordered in a logical way, dealing with what each structure of interest can say about (mostly) secondary adaptation to life in water. However, it seems as though there are two main reasons for increased blood flow in metriorhynchids under discussion here – extra blood for salt excretion, or for some sort of thermoregulation. I can’t help thinking that the discussion could be made shorter by dealing with these (and other adaptations) and bringing in evidence from the osteology to support or refute them. However, I am not suggesting restructuring the section, just that the story suggested by the title could have been made clearer and stronger this way.

There should, however, be some mention of other possibilities for increased blood volume in the skull in the pelagic taxa. One obvious one would be as a cushion for delicate neural tissue for rapid barometric gradients experienced during diving (if, indeed, these animals were diving in any significant way). This would also account for the poorer fidelity of the brain cavity endocast – a larger ‘jacket’ of sinusoidal tissue. Another relates to brain size (large brains need more blood), but this can easily be discounted as a variable.
There is a mix of anatomical directional terms in the text (e.g., anterior/rostral; posterior/caudal). This needs to be standardised.

·

Basic reporting

The manuscript describes the braincase anatomy of two thalattosuchian taxa based on CT scans. The descriptions are much-needed additions to the body of literature on the neural anatomy of the group. The authors demonstrate a robust knowledge of the topic and the results will improve our understanding of the evolution of the group. This manuscript will certainly contribute a lot to future analyses of the group and of the general crocodyliform evolution.

The manuscript is well-structured, although there is room for improvement.

1) The english used is sometimes difficult to understand. I did some language corrections and I tried my best to suggest changes without significantly altering the text, but I would still recommend the authors send the manuscript for a language review.

2) The description part of the manuscript bears comparisons that sometimes make the text a bit confusing. The authors should consider moving the comparisons to another section, to avoid repetition and to make the descriptions more objective. This include the endocast as well as the osteological comparisons.

3) The figures are small. Maybe this is an issue of the PeerJ platform, but I fear that even taking into account the proportion of the abbreviations and the figures themselves, that the space could be better used.

4) Figure callings and figure abbreviations are sometimes not compatible. For instance, when the text refers to the structure X, many times structure X is not pointed at in the figure. There are many instances where the same structure is pointed at in different views. This number could be reduced in order to make room for other relevant structures. The same features need not be marked all the time.

5) In some cases, instead of marking structures on the photos, I suggest a line drawing is provided together with the specimen photo, and that these markings be left to the line drawings only.

Experimental design

CT scanning was the most appropriate approach for this manuscript. The scanning of fossil braincase has yielded a lot of information on the internal braincase anatomy of several fossil taxa, but many groups are still undersampled. This is the case of thalattosuchians. Several aspects of their anatomy remain dubious and this manuscript adds to the body of literature coving this topic.

There are two simple analyses, however, that could enrich the manuscript.

1) Ancestral State Reconstruction. With this simple analysis of the software Mesquite, it is possible to reconstruct character states for internal branches based on the scorings of the terminal taxa. It will make the statements of the manuscript more robust.

2) Time Calibration. This analysis may slightly run out of the scope of this work, but the authors express their wish of it at the end of the manuscript and it's not too time-consuming. It would put the results in a nice geochronological perspective.

Validity of the findings

The anatomical analyses, although not too extensive, expands our anatomical knowledge to several thalattosuchian groups. The paleobiological implications are tackled for the whole group in a much broader sense.

This covers most of the fidings reported in the manuscript. However, the section "Braincase and endocranial anatomy evolution of thalattosuchians" cannot the sustained without a more robust analysis, as suggested in the previous topic (the Ancestral State Reconstruction).

Depending on the number of taxa, the character states of the internal branches are not too difficult to be determined by hand, but being such a simple computational analysis, I would strongly recommend adding the ASR to the manuscript in order to better support the hypotheses erected here.

Additional comments

I congratulate the authors for this manuscript. The descriptions are much-needed additions to the body of knowledge on the thalattosuchian braincase anatomy. CT scanning is much needed in thalattosuchian literature. The group is overall undersampled and such information can contribute a lot to future analyses of the group and of the general crocodyliform evolution.

I have made only a few suggestions regarding manuscript structure and form. However, I highly recommend adding an Ancestral State Reconstruction analysis to the study in order to better support the hypotheses on thalattosuchian braincase anatomy. It is not time-consuming and will add robustness to the results.

The suggestions made here are only general ones. Please check the annotated PDF to address more particular issues.

·

Basic reporting

no comment

Experimental design

no comment

Validity of the findings

no comment

Additional comments

Herrera et al. describe the braincases, endocasts, and semicircular canals of two distantly related thalattosuchian specimens. They provide important new data on the neuroanatomical features of this clade and discuss how it may relate to their adaptation to the marine realm. This is a well-done study and well-written paper. I think it only needs some minor changes to be ready for publication. I very much enjoyed reading this (especially in comparison to other descriptive papers I’ve reviewed recently) and feel it will make a valuable contribution to the growing literature on crocodylomorph neuroanatomical evolution.

Comments:

I received a word document rather than a pdf for review, so I made most of my comments as “track changes” annotations in Word. I also made a number of minor English language spelling/grammar corrections, but as I said above, it was well written, so these aren’t voluminous.

In general, I think the presence of a single XII foramen as characteristic of thalattosuchians is a bit overstated in this manuscript. While most may have only a single opening, some have multiple. More importantly, a single XII opening is fairly common among crocodyliforms, and where we have good sampling, seems fairly variable within species. The authors are welcome to keep this in the manuscript as most thalattosuchians do only have a single opening, however, they may wish to emphasize that this is not unique to thalattosuchians.

Comments on figures:
I did not receive any figure captions with my review materials, so I could not evaluate these.

Could the authors add a white shadow behind the label lines on the endocast figures (Figs. 3, 4, 7, &8) as they did for figures 1, 5, &6? The lines are hard to follow across the darker colors.

Fig 6C – could the authors add a dashed line for the groove for V2 (as they did for V1)?

---

## Round 0.2 · accepted · Accept

Unfortunately Reviewer 1 was unavailable to re-review the manuscript, however both Reviewers 2 and 3 have evaluated it and are recommending Acceptance.

Therefore, I am happy to Accept this submission.

# ·

Basic reporting

This submission is about the reviewed manuscript on the braincase anatomy of two thalattosuchian taxa based on CT scans. It is a very welcome addition to the literature on the neural anatomy of the group.

The authors have addressed all the reviews accordingly. They accepted most of modifications and while they refused others, these were all well justified. I accept them and have no further issues with the manuscript.

Among the accepted modifications are new figures, including line drawings and a cladogram with character states plotted on it. These are very well done and they have certianly improved the quality of the manuscript, helping clarify some points and guiding the reader more easily through the text.

This manuscript will contribute to expanding our knowledge in crocodyliform evolution and therefore I recommend the manuscript for publication.

Experimental design

Nothing to add.

Validity of the findings

Nothing to add.

·

Basic reporting

no comment

Experimental design

no comment

Validity of the findings

no comment

Additional comments

I am well satisfied with the revisions that the authors have made to the manuscript and the manuscript and figures are much improved. I think they did a good job of addressing the issues raised by the reviewers and I feel the paper is basically ready for publication. There are still a few lingering grammatical issues, but they are much fewer and can be cleaned up during the editing/proofing stage. Congrats to the authors.